# MOSDT: Self-Distillation-Based Decision Transformer for Multi-Agent Offline Safe Reinforcement Learning

**Yuchen Xia**[1]    **Yunjian Xu**[1*]
[1]The Chinese University of Hong Kong
ycxia@link.cuhk.edu.hk, yjxu@mae.cuhk.edu.hk

## Abstract

We introduce MOSDT, the first algorithm designed for multi-agent offline safe reinforcement learning (MOSRL), alongside MOSDB, the first dataset and benchmark for this domain. Different from most existing knowledge distillation-based multi-agent RL methods, we propose policy self-distillation (PSD) with a new global information reconstruction scheme by fusing the observation features of all agents, streamlining training and improving parameter efficiency. We adopt full parameter sharing across agents, significantly slashing parameter count and boosting returns up to 38.4-fold by stabilizing training. We propose a new plug-and-play cost binary embedding (CBE) module, which binarizes cumulative costs as safety signals and embeds the signals into return features for efficient information aggregation. On the strong MOSDB benchmark, MOSDT achieves state-of-the-art (SOTA) returns in 14 out of 18 tasks (across all base environments including *MuJoCo*, *Safety Gym*, and *Isaac Gym*) while ensuring complete safety, with only $65\%$ of the execution parameter count of a SOTA single-agent offline safe RL method CDT. Code, dataset, and results are available at this website: https://github.com/Lucian1115/MOSDT.git

## 1 Introduction

Offline reinforcement learning (Offline RL) leverages static datasets to derive policies [1]. Its applications span diverse domains, such as large language models (LLMs) [2], robotics [3], and power systems [4]. Offline RL has spawned specialized subfields, including offline multi-agent RL (offline MARL) [5, 6] for systems with interacting agents, and offline safe RL [7, 8] for safety-critical problems. However, multi-agent offline safe RL (MOSRL) remains largely unstudied. MOSRL seeks to learn safe policies for multiple agents in an interaction-free way, offering significant potential for distributed safety-critical applications, like autonomous vehicle coordination [9], power grid scheduling [10], and robot collaboration [11]. While existing offline RL methods provide a solid foundation, they may not fully address the complexities of multi-agent cooperation, offline learning, and safety assurance within such contexts.

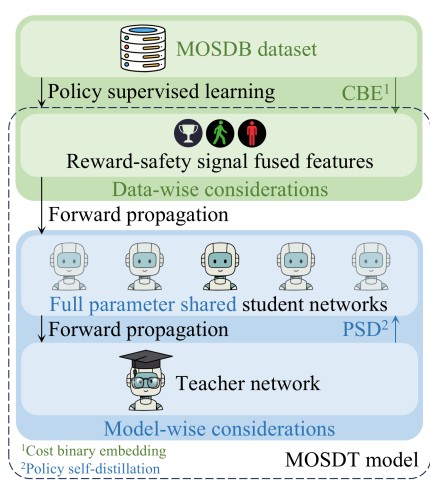

Figure 1: Training MOSDT on MOSDB.

Addressing MOSRL requires tackling the key challenges from both offline MARL and offline safe RL. For offline MARL, the centralized training with decentralized execution (CTDE) framework has emerged as

---

*corresponding author

39th Conference on Neural Information Processing Systems (NeurIPS 2025).

a prevalent communication-free solution. Within this paradigm, knowledge distillation (KD) [12]-based methods (like MADTKD) [5] show strong performance by training a teacher network and then distilling policy from the teacher to students. Self-distillation (SD) [13] is a streamlined KD variant with teacher-student integration and the synchronization of supervised learning and distillation. SD is well-suited for multi-agent applications due to its efficiency. Despite the success of SD in computer vision [13, 14, 15], its effectiveness in MARL remains untested. Besides, MARL methods generally suffer from training instability resulting from dynamic policy improvements among different agents [16, 17].

In offline safe RL, decision transformer (DT) [18]-based methods, such as CDT [7], achieve notable performance by leveraging the causal transformer network [19]. These approaches need targets for returns and cumulative costs to launch the execution. Existing methods typically set cost targets as specific values (like predetermined thresholds [7]), converging actual cumulative costs to the predetermined values [20, 21]. Unfortunately, such convergence may lead to suboptimal and/or potentially unsafe actions along trajectories. In addition, as causal transformers can only attend to preceding tokens, they would overlook the cumulative cost or the return, specifically whichever occurs later in the input sequence.

This work proposes MOSDT (multi-agent offline safe decision transformer), the *first* algorithm tailored to MOSRL. Fig. 1 shows the process of training MOSDT on our proposed MOSDB dataset. To streamline the training process and the network structure, we propose policy self-distillation (PSD) to build an efficient CTDE framework instead of adopting the two-stage conventional KD (used in MADTKD [5]). For MARL, PSD distinguishes from SD [13] with a new global information reconstruction scheme by summing up the observation features of all agents. PSD integrates student networks within the teacher network and is performed synchronously with policy supervised learning—a pioneering design reducing training parameter count by $35\%$ and training time by $24\%$. MOSDT marks the first demonstration that SD is effective in MARL, hinting at broader applicability for existing methods.

To ensure training stability, we adopt full parameter sharing across all agents, while other offline MARL methods with sharing designs [22, 23] share a subset of parameters. Compared to the case with no/partial parameter sharing, full parameter sharing significantly boosts returns for tasks with three or more agents (in *MuJoCo* environment [24]): 10.1 to 38.4-fold for a three-agent task, 2.2 to 2.3-fold for a four-agent task, and 1.6 to 5.6-fold for a six-agent task. Full parameter sharing significantly slashes training parameter count by $47\%$ and execution parameter count by $58\%$, making MOSDT much more lightweight and scalable.

To improve existing cost processing methods, we propose cost binary embedding (CBE), a plug-and-play module consisting of cost binarization and safety signal embedding. CBE fuzzifies cost targets to prevent actual cumulative costs from converging to predetermined values in execution. Towards this goal, we binarize cumulative costs as safety signals during training. We embedded the safety signals into returns, explicitly passing reward-cost correlations to causal transformers for sharper information aggregation. While ensuring safety, CBE improves returns on 14 out of 18 tasks (across all base environments, including *MuJoCo* [24], *Safety Gym* [25], and *Isaac Gym* [26]) in MOSDT.

Finally, we build the MOSDB dataset by collecting training data from 2 online algorithms [27] on all safe MARL tasks in Safety Gymnasium [28], following the online-collection paradigm in DSRL [29] (a widely used single-agent safe RL dataset). Given the absence of direct baselines for MOSRL, we follow [30] to compare MOSDT with centralized single-agent offline safe RL methods on MOSDB. Extensive experiments demonstrate the state-of-the-art (SOTA) performance of MOSDT. Ablation experiments validate the contributions of PSD, full parameter sharing, and CBE to return enhancement and risk control.

Our contributions are: (1) MOSDT, the first MOSRL algorithm, achieving high parameter efficiency and a streamlined training process by PSD. (2) A full parameter sharing design that stabilizes training and makes MOSDT lightweight and scalable. (3) CBE, A plug-and-play module, offering an intuitive alternative to specific cost targets and improving information aggregation. (4) MOSDB, the first MOSRL dataset and benchmark, on which MOSDT achieves SOTA performance in 14 out of 18 tasks with only $65\%$ of the execution parameter count of our base model CDT [7].

## 2 Related work

**Offline RL**  DT [18] proposes a powerful paradigm that formulates offline RL as a sequence modeling problem, inspiring subsequent research. For example, MGDT [31] achieves rapid adaptation to new tasks through fine-tuning. EDT [32] supports trajectory stitching by adjusting history lengths. HDT [33] leverages subgoal states to make decisions. However, the performance of DT in MOSRL remains unstudied.

**Offline MARL**  For multi-agent systems, the CTDE framework eliminates information dependencies between training and execution, bypassing the communication requirements posed by partial observability. Recent works [5, 34, 35] adopt KD to implement the CTDE framework. MADTKD [5] aligns the output between partially observable student DTs and a globally observable teacher DT. PTDE [34] tailors the global information to each agent and distills it into partial observations. LDPD [35] uses LLMs as teachers to train compact students. The two-stage training of KD may lead to scalability issues in MARL [5]. SD [13] offers a streamlined alternative to KD, although its suitability for MARL remains unknown. Additionally, while existing offline MARL methods at most partially share parameters across agents [22, 23], full parameter sharing has not been attempted.

**Offline safe RL**  DT-based methods again show impressive performance. For example, CDT [7] sends augmented data into a causal transformer in the order of returns, cumulative costs, and observations to predict actions. Saformer [21] leverages cost-related tokens and posterior safety checks to craft constraint-compliant policies. SDT [20] employs signal temporal logic to define time-sensitive safety rules for agents. Despite these strides, predetermined cost targets would make actual cumulative costs converge to predetermined targets. Moreover, due to the masked attention mechanism, causal transformers inherently overlook either the cumulative cost or the return—whichever appears later in the sequence.

Most offline safe RL algorithms are trained on the DSRL dataset [29]. It is gathered from environments like Safety Gymnasium [28]. However, there is a lack of datasets tailored for MOSRL.

## 3 Methods

Following CDT [7], we formulate MOSRL as a probability distribution learning problem, in which agent $i$ ($i = 1, \ldots, N$ and $N$ is the number of agents) uses offline data to learn a safe policy $\pi^i : \mathcal{S}^i \times \mathcal{A}^i \to [0, 1]$, where $\mathcal{S}^i$ is the state space and $\mathcal{A}^i$ is the action space.

For MOSRL, we propose the first algorithm, MOSDT. Following CDT [7], we adopt DT as the base network. Inspired by MADTKD [5], we also build the CTDE architecture through distillation. Specifically, for agent $i$ at time $t$, in centralized training, a teacher network $\pi^i_T$ regresses its action $\hat{a}^i_t$ from a historical trajectory $\mathcal{T}_t = (R_1, C_1, o_1, a_1, \ldots, R_t, C_t, o_t)$, containing the global information preceding $a^i_t$. $R_t = \sum_{\tau=t}^{\mathcal{M}} r_\tau$ is the system reward return ($r_\tau$ is the system reward at time $\tau$, and $\mathcal{M}$ is the maximum episode length). $C_t = \sum_{\tau=t}^{\mathcal{M}} c_\tau$ is the system cumulative cost ($c_\tau$ is the system cost at time $\tau$). $o_t$ is the global observation, and $a_t$ is the joint action ground truth. In decentralized execution, $\hat{a}^i_t$ is predicted by a student network $\pi^i_S$ from partial information $\mathcal{T}^i_t = \left( R^i_1, C^i_1, o^i_1, a^i_1, \ldots, R^i_t, C^i_t, o^i_t \right)$. For the tasks where environments offer only system rewards and costs (cf. Table 13 in Appendix C), we set $R^i_t = R_t$ and $C^i_t = C_t$.

### 3.1 CTDE via policy self-distillation (PSD)

Given the intrinsic applicability of SD [13] for MARL, we propose to utilize it to build the CTDE architecture. Student outputs, teacher outputs, and ground truths are aligned by three synchronous processes during training: decentralized student supervised learning (DSSL), centralized teacher supervised learning (CTSL), and policy self-distillation (PSD). Fig. 2 illustrates the network structure of MOSDT.

**Decentralized student supervised learning (DSSL)**  Each agent adopts an individual DSSL module based on CDT [7] to directly align the policy to ground truths. We propose to adopt full parameter sharing across all DSSLs (as detailed in Section 3.2). For agent $i$ at time $t$, the details of DSSL

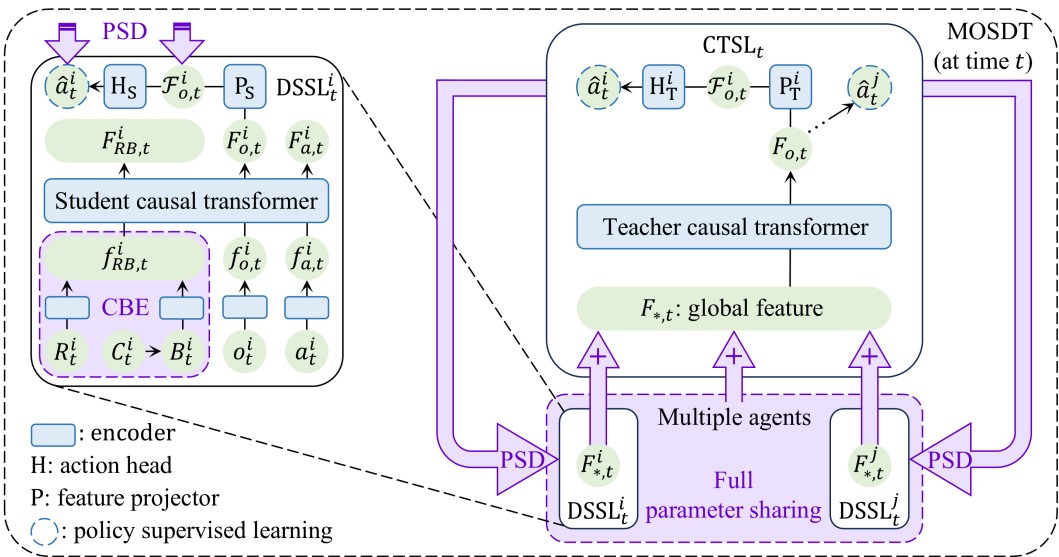

Figure 2: The network structure of the proposed MOSDT algorithm. The purple areas indicate our proposed innovative modules. The green areas represent data, while the blue areas represent networks. We enlarge one of the DSSL modules and display it in detail in the upper left corner. We only illustrate the situation at time $t$.

are shown in the "DSSL$_t^i$" box in Fig. 2. Each input is encoded into a feature representation by a distinct encoder. In this encoding process, our proposed CBE module binarizes $C_t^i$ as a safety signal $B_t^i$ and obtains a reward-safety signal fused feature $f_{RB,t}^i$ (as detailed in Section 3.3). The encoded features $f_{*,t}^i$ ($* \in \{RB, o, a\}$) are refined by a student causal transformer. We use $F_{*,t}^i$ to denote the refined features. $\hat{a}_t^i$ is generated by a student action head $H_S^i$ from the refined observation feature $F_{o,t}^i$. Following MADTKD [5], we insert a student feature projector $P_S^i$ before $H_S^i$ to perform feature distillation.

Supervised learning for student policies (the blue dashed circles at the top-left of Fig. 2) minimizes the following loss function:

$$Loss_S^i = \frac{1}{M} \sum_{t=1}^{M} -\log \pi_S^i \left( a_t^i \mid \mathcal{T}_t^i \right) - \lambda_S^i H \left( \pi_S^i \right), \tag{1}$$

where $M$ is the training sequence length, and $H \left( \pi_S^i \right)$ weighted by learnable $\lambda_S^i$ is a widely used entropy regularization term [36].

**Centralized teacher supervised learning (CTSL)** CTSL directly aligns teacher outputs with ground truths. At time $t$, the details of CTSL are shown in the "CTSL$_t$" box in Fig. 2. Inspired by the success of SD in computer vision [13, 14, 15], we integrate student networks into the teacher network for decision making. To this end, we propose a new global information reconstruction scheme for SD in MARL, by summing up $F_*^i$ across all agents to gain a global feature:

$$F_{*,t} = \sum_{i=1}^{N} F_{*,t}^i, \tag{2}$$

as indicated by the purple straight arrows in Fig. 2. Similar to the DSSL module, a teacher causal transformer and a teacher action head $H_T^i$ (with a teacher feature projector $P_T^i$) are used to map the global feature $F_{*,t}$ into the action output $\hat{a}_t^i$. The teacher causal transformer is shared across agents, while $P_T^i$ and $H_T^i$ possess distinct parameters for each individual $i$.

We keep $F_{*,t}^i$ in the computational graph of Eq. (2) instead of detaching it to get better results (cf. Table 10 in Appendix B.5). Through Eq. (2), students serve as components of the teacher for teacher-student policy integration, and the teacher is able to access global information $\mathcal{T}_t$, achieving a centralized teacher training.

Supervised learning for the teacher policy (the blue dashed circles in the "CTSL$_t$" box in Fig. 2) minimizes the following loss function:

$$Loss_T^i = \frac{1}{M} \sum_{t=1}^{M} -\log \pi_T^i \left( a_t^i \mid \mathcal{T}_t \right) - \lambda_T^i H \left( \pi_T^i \right), \tag{3}$$

where $\lambda_T^i$ is the learnable weight for entropy regularization.

In execution, we only use student policies $\pi_S^i$ to predict actions $\hat{a}_t^i$ from partial information $\mathcal{T}_t^i$, decentralizing the execution.

**Policy self-distillation (PSD)** PSD aligns student outputs with teacher outputs, achieving a novel integration of SD and MARL. We bridge the performance gap between partially observable students and the teacher with global information access by aligning both their actions and feature representations. We minimize the following loss function:

$$Loss_D^i = w_A D_{KL} \left( \pi_S^i \mid \pi_T^i \right) + w_F \left\| \mathcal{F}_{o,S}^i - \mathcal{F}_{o,T}^i \right\|_2, \tag{4}$$

as shown by the two purple curved arrows with "PSD" in Fig. 2. $D_{KL} (* \mid *)$ is the KL divergence. $\mathcal{F}_{o,S}^i$ is the whole observation feature generated by the student feature projector $P_S^i$, and $\mathcal{F}_{o,T}^i$ is the whole observation feature generated by the teacher feature projector $P_T^i$. $\|*\|_2$ is the Euclidean norm. $w_A$ and $w_F$ are weights that indicate the importance of action and feature distillation, respectively (both are set to 0.5 with reference to MADTKD [5]). In Eq. (4), the teacher policy $\pi_T^i$ and the teacher projected feature $\mathcal{F}_{o,T}^i$ are detached from the computational graph so that PSD only updates student policies.

Following SD [13], we carry out PSD synchronously with DSSL and CTSL to streamline the training process. Specifically, we minimize the following total loss function:

$$Loss = \sum_{i=1}^{N} Loss_S^i + Loss_T^i + Loss_D^i. \tag{5}$$

All loss functions are averaged across samples in a mini-batch.

**Network details** Network details are consistent with those in CDT [7] if available. The data augmentation in CDT [7] is also adopted in MOSDT with the same settings. We use linear layers as the encoders. Timestamp embeddings are added to the encoded features to incorporate temporal information. Feature projectors are linear layers followed by GELU activation functions [37]. Each action head consists of 2 linear layers: one for computing mean values of action distributions and another for standard deviations (the latter is not used in execution). More network details of MOSDT are summarized in Appendix A.1. The algorithms of the training and execution processes of MOSDT are presented in Appendix A.3.

## 3.2 Full parameter sharing among agents

To tackle the training instability problem caused by dynamic policy improvements among different agents and to improve model scalability in offline MARL, we propose to adopt full parameter sharing across all agents, for the first time. Specifically, each student shares identical parameters, as expressed in the following equation:

$$\pi_S^1 = \cdots = \pi_S^N, \tag{6}$$

implying the equivalence of each network component across students throughout the training process. In practice, we only create one instance of the Student class for all agents.

In the entire MOSDT network, only the teacher action heads and the teacher feature projectors are distinct for different agents.

## 3.3 Cost binary embedding (CBE)

Cost binary embedding (CBE) is a straightforward and plug-and-play method (the purple dashed "CBE" box in Fig. 2) that provides an intuitive alternative to specific cost targets and enhances information aggregation. CBE consists of cost binarization and safety signal embedding.

**Cost binarization** In DT-based offline safe RL methods [7, 21, 20], at the beginning of execution ($t = 1$), $R_1$ and $C_1$ need to be assigned as targets. Most existing algorithms tend to generate trajectories with cumulative costs approaching the predetermined targets, leading to suboptimal and/or potentially unsafe actions along trajectories.

To tackle this issue, CBE applies fuzzification to cost targets, preventing the cumulative costs along actual trajectories from approaching predetermined specific values during execution. Towards this goal, in training we map $C_t^i$ as a safety signal:

$$B_t^i = \begin{cases} 0 & \text{if } C_t^i \leq c, \\ 1 & \text{otherwise,} \end{cases} \tag{7}$$

where $c$ is the cost threshold (often fixed). "0" represents "safe", and "1" represents "unsafe". After training MOSDT with Eq. (8), we input $B_t^i = 0$ to the model throughout execution, guiding the model to yield safe trajectories. ($R_1^i$ is set to a task-related large value as CDT [7]).

**Safety signal embedding** CDT [7] separately processes $R_t^i$ and $C_t^i$ by a causal transformer, which prevents the model from capturing their correlations during the processing of $R_t^i$ until $C_t^i$ starts being processed, due to the masked attention mechanism. To allow MOSDT to always access the prior information about the reward-cost correlation, we embed the safety signal feature $f_{B,t}^i$ into the return feature $f_{R,t}^i$ by concatenating them together. By slight abuse of notation, we use $f_{RB,t}^i$ to denote the fused feature:

$$f_{RB,t}^i = \text{concatenate}\left(f_{R,t}^i, f_{B,t}^i\right). \tag{8}$$

$f_{R,t}^i$ and $f_{B,t}^i$ are of 64 dimensions, half the size of $f_{o,t}^i$ and $f_{a,t}^i$. Their concatenation results in a 128-dimensional $f_{RB,t}^i$, ensuring the dimensional consistency throughout MOSDT.

### 3.4 MOSDB dataset

We introduce MOSDB, the first dataset and benchmark for MOSRL. To obtain diverse trajectory data, we train 2 online safe MARL methods, MACPO and MAPPO-Lagrangian [27], across all 18 safe MARL tasks in Safety Gymnasium [28] (except for tasks in "Goal 0" series with constantly zero cost). For each task, training was conducted with 3 cost thresholds: 25 (the default), 15, and 5 (two lower values selected to ensure dense trajectory distributions in sub-threshold regions). All other training hyperparameters retained their default settings. The collected trajectory data is filtered by the density filter in the DSRL dataset [29] to maintain suitable densities over return-cumulative cost planes (cf. Fig. 5 in Appendix C). MOSDB dataset comprises approximately $2 \times 10^7$ data tuples, occupying 18.7 GB on Linux.

The MOSDB benchmark contains 3 task sets:

- MOS Velocity. Robots are required to move as quickly as possible while adhering to velocity constraints. Multiple agents need to control distinct body segments cooperatively.

- MOS Goal. Each agent is required to reach its color-designated target while avoiding collisions and hazardous terrain.

- MOS Isaac Gym. It focuses on collaborative robotic tasks, such as coordinated ball-handovers between dual manipulators, with enforced safety constraints on joint movements.

## 4 Experiments

We use PyTorch to program on Linux. Experiments are conducted at an NVIDIA® GeForce RTX™ 4090 D GPU (24 GB VRAM) with an AMD® Ryzen™ 9 7950X CPU.

### 4.1 MOSDB benchmark and the performance of MOSDT

We use the MOSDB dataset to train MOSDT and all single-agent offline safe RL methods provided by the DSRL dataset [29]:

- BC [7]: Behavior cloning.

Table 1: MOSDB benchmark and the performance of MOSDT. Results are in the "return (cumulative cost)" format. The cost threshold $c$ is 25 (consistent with the original online tasks), and the maximum return of each task is shown in Table 13 in Appendix C. Blue: Safe policies with the highest rewards. Red: Unsafe policies. Due to space constraints, the sample standard deviation across multiple runs of each experiment is shown in Table 11 in Appendix B.6.

| Task | BC [7] | BCQ-Lag [8] | BEAR-Lag [8] | CDT [7] | COptiDICE [41] | CPQ [8] | MOSDT (ours) |
|---|---|---|---|---|---|---|---|
| **MOS Velocity** | | | | | | | |
| 2x1Swimmer | 5.08 (7.27) | 8.13 (11.53) | 5.24 (9.33) | 9.25 (12.70) | 0.97 (12.13) | 8.20 (13.30) | 11.64 (20.33) |
| 2x3HalfCheetah | 2043.94 (22.83) | 2162.67 (62.43) | 2181.00 (57.90) | 2087.33 (40.03) | 2029.00 (10.03) | 406.90 (8.97) | 2052.64 (22.27) |
| 2x3Walker2d | 1512.75 (0.00) | 1578.59 (10.17) | 1515.60 (3.53) | 1526.43 (3.67) | 1540.41 (0.00) | 315.20 (15.97) | 1584.87 (3.83) |
| 2x4Ant | 2361.63 (0.00) | 2472.33 (6.77) | 2217.89 (0.90) | 2116.81 (7.77) | 2125.14 (1.67) | -1488.45 (0.50) | 2054.88 (0.87) |
| 3x1Hopper | 31.63 (0.30) | 40.20 (1.83) | 82.24 (8.33) | 27.15 (0.00) | 69.02 (7.40) | 121.58 (1.07) | 1122.23 (4.00) |
| 4x2Ant | 831.82 (0.00) | 779.85 (0.00) | 792.78 (0.00) | 923.72 (0.00) | 816.86 (0.00) | -466.82 (0.83) | 2083.85 (3.53) |
| 6x1HalfCheetah | 447.13 (0.00) | 339.63 (0.03) | 334.62 (0.00) | 397.80 (0.00) | 321.46 (0.00) | -201.40 (0.87) | 1853.64 (21.97) |
| 9|8Humanoid | 575.73 (20.70) | 581.42 (19.27) | 571.16 (18.57) | 545.80 (20.87) | 554.40 (15.27) | 404.48 (18.53) | 444.71 (22.80) |
| **MOS Goal** | | | | | | | |
| Multi-Ant1 | 23.41 (17.00) | 15.78 (23.85) | 21.31 (9.50) | 28.94 (10.67) | 33.75 (14.00) | 0.61 (0.00) | 38.38 (14.50) |
| Multi-Ant2 | 2.59 (8.00) | 1.90 (21.05) | 2.71 (16.17) | 4.93 (17.50) | 3.55 (17.33) | 0.51 (0.00) | 2.96 (7.50) |
| Multi-Point1 | 6.47 (17.17) | -1.43 (25.48) | 4.28 (7.67) | 9.25 (14.83) | 2.11 (8.17) | 2.84 (7.67) | 9.65 (12.67) |
| Multi-Point2 | -0.06 (4.33) | -8.59 (36.53) | 1.20 (14.00) | 3.07 (19.33) | -1.23 (18.83) | 0.95 (9.67) | -1.08 (21.00) |
| **MOS Isaac Gym** | | | | | | | |
| CloseDrawerMA | -5.23 (1.00) | -5.15 (0.80) | -4.49 (3.57) | -3.45 (0.00) | -5.28 (0.00) | -5.43 (0.00) | -3.45 (0.00) |
| PickAndPlaceMA | -5.72 (3.73) | -4.92 (0.27) | -4.26 (0.00) | -5.50 (0.00) | -7.58 (2.67) | -5.32 (2.47) | -2.67 (0.00) |
| CatchFingerMA | 0.19 (0.00) | 0.20 (2.60) | 0.23 (0.00) | 0.18 (7.60) | 0.22 (0.00) | 0.11 (6.23) | 0.25 (6.33) |
| CatchJointMA | 0.21 (0.00) | 0.19 (0.00) | 0.20 (0.00) | 0.26 (0.80) | 0.24 (0.00) | 0.15 (5.97) | 0.31 (0.00) |
| OverFingerMA | 0.43 (0.00) | 0.46 (0.07) | 0.44 (0.00) | 0.52 (0.54) | 0.44 (0.00) | 0.39 (8.00) | 0.52 (0.00) |
| OverJointMA | 0.45 (0.00) | 0.47 (0.00) | 0.46 (0.00) | 0.46 (5.63) | 0.44 (0.00) | 0.42 (8.00) | 0.47 (1.03) |
| Summary | 0 SOTA (safe) | 3 SOTA (unsafe) | 0 SOTA (unsafe) | 4 SOTA (unsafe) | 0 SOTA (safe) | 0 SOTA (safe) | 14 SOTA (safe) |

- BC-Safe [7]: Behavior cloning using only safe trajectories (results are presented in Table 3 in Appendix B.1).
- BCQ-Lag [8]: A BCQ [38]-based method incorporating cost thresholds with PID-Lagrangian [39].
- BEAR-Lag [8]: A BEAR [40]-based method that deals with safety constraints by PID-Lagrangian [39].
- CDT [7]: A DT [18]-based method that sends augmented data into a causal transformer in the order of returns, cumulative costs, and observations to predict actions.
- COptiDICE [41]: A OptiDICE [42]-based method using the Lagrangian approach to maintain safety.
- CPQ [8]: A constrained Q-updating method that incorporates penalties for unseen or unsafe actions.

Baseline algorithms are trained within the CTCE framework, accessing global information and thus yielding better performance. All methods are trained for $10^5$ steps, with evaluations conducted at 40 checkpoints during training. Each evaluation is averaged over 10 interaction episodes with Safety Gymnasium [28]. Experiments are repeated across 3 fixed random seeds. All above hyperparameters are consistent with those in the DSRL dataset [29]. More training and evaluation settings are presented in Appendix A.2. We report the highest returns achieved under the cost threshold as final performance. For unsafe policies, we report the returns with the lowest cumulative costs. Table 1 summarizes the MOSDB benchmark and the performance of MOSDT.

While the strong baseline methods leverage global information, MOSDT utilizes only partial information and achieves the highest returns on 14 out of 18 tasks (across all base environments including *MuJoCo* [24], *Safety Gym* [25], and *Isaac Gym* [26]) while guaranteeing safety on all tasks, demonstrating its outstanding capability in balancing return maximization and risk control. MOSDT delivers very strong performance in tasks with more than three agents, "3x1Hopper", "4x2Ant", and "6x1HalfCheetah", indicating a remarkable capability to address multi-agent challenges. MOSDT attains SOTA returns across all challenging "MOS Isaac Gym" tasks with high-dimensional action spaces (cf. Table 13 in Appendix C), demonstrating its superiority in making complex decisions.

We evaluate the parameter efficiency of MOSDT against its base model CDT [7] by averaging their total/execution parameter counts across all tasks. The parameter counts of CDT [7], MOSDT, and MOSDT variants are presented in Fig. 3. Due to the lightweight student network design and the full parameter sharing, MOSDT requires only 65% of the execution parameter count of CDT [7] while

Table 2: Ablation study results. Results are in the "return (cumulative cost)" format. The cost threshold $c$ is 25 (consistent with the original online tasks). The maximum return of each task is shown in Table 13 in Appendix C. "⇓": Policies with lower returns than full MOSDT or unsafe policies. Red: Unsafe policies. "Partial PS": Without parameter sharing for student action heads. "No PS": No parameter sharing. "w/o SE": Without safety signal embedding. Due to space constraints, the sample standard deviation across multiple runs of each experiment is shown in Table 12 in Appendix B.6.

| Task | MOSDT | w/o PSD | Partial PS | No PS | w/o SE | w/o CBE |
|---|---|---|---|---|---|---|
| **MOS Velocity** | | | | | | |
| 2x1Swimmer | 11.64 (20.33) | ⇓ 11.58 (9.40) | ⇓ 10.97 (15.33) | ⇓ 10.32 (4.20) | ⇓ 10.40 (16.80) | ⇓ 11.08 (23.00) |
| 2x3HalfCheetah | 2052.64 (22.27) | ⇓ 2081.00 (39.83) | ⇓ 2050.14 (23.00) | ⇓ 2013.23 (22.13) | ⇓ 2034.48 (22.03) | ⇓ 2009.30 (20.63) |
| 2x3Walker2d | 1584.87 (3.83) | ⇓ 1576.03 (4.23) | 1585.52 (2.07) | 1597.48 (2.03) | ⇓ 1577.04 (2.93) | ⇓ 1568.17 (6.77) |
| 2x4Ant | 2054.88 (0.87) | ⇓ 2033.27 (2.97) | ⇓ 2049.88 (1.73) | 2059.04 (7.90) | ⇓ 2008.38 (6.17) | ⇓ 2023.90 (2.93) |
| 3x1Hopper | 1122.23 (4.00) | 1239.49 (10.67) | ⇓ 110.93 (1.33) | ⇓ 29.26 (0.00) | ⇓ 1115.51 (13.60) | ⇓ 1112.48 (6.03) |
| 4x2Ant | 2083.85 (3.53) | 2189.01 (5.57) | ⇓ 962.59 (0.00) | ⇓ 914.51 (0.00) | 2128.76 (3.43) | ⇓ 2070.38 (8.00) |
| 6x1HalfCheetah | 1853.64 (21.97) | ⇓ 1823.33 (27.27) | ⇓ 1169.26 (0.10) | ⇓ 528.80 (0.00) | ⇓ 1835.89 (22.93) | ⇓ 1823.27 (22.53) |
| 9\|8Humanoid | 444.71 (22.80) | 477.70 (21.50) | 474.87 (21.73) | 482.17 (21.60) | 453.52 (21.67) | 456.21 (21.87) |
| **MOS Goal** | | | | | | |
| Multi-Ant1 | 38.38 (14.50) | ⇓ 37.79 (20.33) | ⇓ 36.87 (10.50) | ⇓ 33.91 (8.83) | 40.13 (19.33) | ⇓ 30.93 (16.50) |
| Multi-Ant2 | 2.96 (7.50) | 5.40 (11.83) | 4.60 (11.67) | ⇓ 2.29 (11.33) | ⇓ 2.03 (1.41) | ⇓ 2.28 (11.83) |
| Multi-Point1 | 9.65 (12.67) | 13.09 (14.67) | 10.42 (14.83) | 14.13 (18.67) | 10.87 (12.33) | ⇓ 6.90 (12.50) |
| Multi-Point2 | -1.08 (21.00) | ⇓ -2.17 (11.50) | 0.33 (7.83) | 1.18 (16.67) | 2.84 (17.00) | ⇓ -2.20 (8.67) |
| **MOS Isaac Gym** | | | | | | |
| CloseDrawerMA | -3.45 (0.00) | ⇓ -3.74 (0.00) | ⇓ -3.46 (0.00) | -3.32 (0.00) | ⇓ -3.49 (0.00) | -3.21 (0.00) |
| PickAndPlaceMA | -2.67 (0.00) | -2.23 (0.00) | -2.24 (0.00) | -2.00 (0.00) | -2.34 (2.67) | ⇓ -2.71 (0.00) |
| CatchFingerMA | 0.25 (6.33) | ⇓ 0.21 (4.40) | ⇓ 0.23 (5.33) | ⇓ 0.20 (3.37) | ⇓ 0.22 (2.87) | ⇓ 0.21 (7.40) |
| CatchJointMA | 0.31 (0.00) | ⇓ 0.23 (0.00) | ⇓ 0.27 (0.00) | ⇓ 0.26 (0.00) | ⇓ 0.22 (0.00) | ⇓ 0.23 (0.00) |
| OverFingerMA | 0.52 (0.00) | ⇓ 0.45 (0.93) | ⇓ 0.48 (0.00) | ⇓ 0.44 (0.00) | ⇓ 0.45 (0.00) | 0.60 (3.53) |
| OverJointMA | 0.47 (1.03) | ⇓ 0.44 (0.30) | ⇓ 0.46 (0.00) | ⇓ 0.45 (0.00) | ⇓ 0.45 (0.00) | 0.56 (5.17) |
| Summary | (safe) | 12 ⇓ (unsafe) | 12 ⇓ (safe) | 11 ⇓ (safe) | 12 ⇓ (safe) | 14 ⇓ (safe) |

achieving better performance. Therefore, MOSDT is well-suited for memory-constrained application scenarios, such as micro-robots.

MOSDT consumes less execution time than CDT [7] owing to the lightweight design of the student network. MOSDT performs an inference for one agent in 0.80 milliseconds on average, compared to 0.86 milliseconds for CDT [7]. Detailed efficiency analysis is presented in Appendix B.2.

## 4.2 Ablation study

We conduct massive ablation experiments to evaluate the effectiveness of each component in MOSDT. The training and evaluation settings are consistent with those in Section 4.1. Table 2 summarizes the ablation study results.

**The effect of PSD**   To analyze the effectiveness of PSD, we remove it by minimizing only Eq. (1) during training, converting MOSDT into a decentralized training with decentralized execution (DTDE) architecture. The "w/o PSD" column of Table 2 shows the critical role of PSD. Its absence degrades returns on 12 out of 18 tasks and renders the model unsafe. Given that safety is a fundamental requirement in MOSRL, PSD is vital for MOSDT.

We further verify the effectiveness of the action distillation and the feature distillation (the two terms of Eq. (4), finding that removing either impairs performance (cf. Table 6 in Appendix B.3).

To analyze the efficiency improvements provided by PSD, we train MOSDT by the conventional KD used

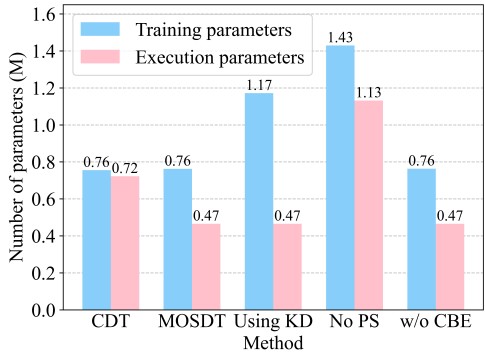

Figure 3: Number of parameters. "Using KD": Using conventional KD instead of PSD. "No PS": No parameter sharing. MOSDT only requires 65% of the execution parameter count of CDT. PSD reduces the training/execution parameter counts by 47%/58%.

in MADTKD [5] for comparison. In this setting, policy distillation is performed after the completion of centralized teacher training. The comparison between the "MOSDT" column and "Using KD"

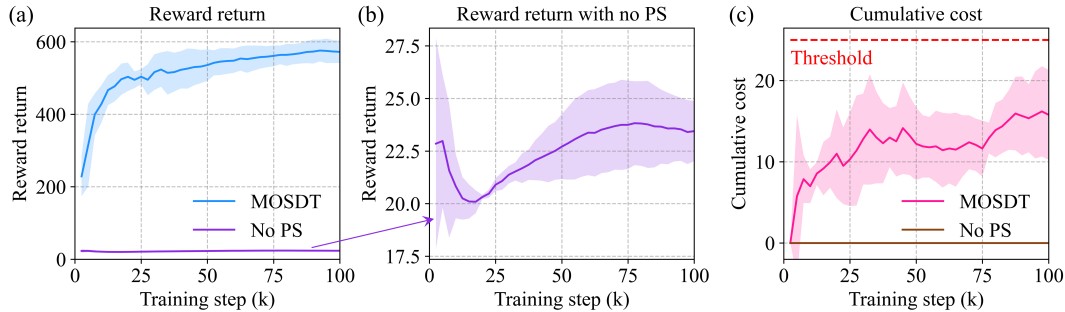

Figure 4: Visualization of the training on the "3x1Hopper" task. "No PS": No parameter sharing. The purple curve in Subgraph (a) is enlarged and displayed in detail in Subgraph (b) for better viewing. Shaded areas represent sample standard deviations across multiple runs. We adopt the smoothing in TensorBoard to better view time-series data trends.

column in Fig. 3 reveals that PSD reduces the number of training parameters by $35\%$ compared to using KD, due to the proposed teacher-student policy integration. In addition, PSD decreases the average training time by $24\%$ (cf. Table 5 in Appendix B.2) through the synchronization of policy supervised learning and policy distillation.

**The effect of full parameter sharing** To verify the effects of full parameter sharing, we train MOSDT with no parameter sharing. The Comparison between the "MOSDT" column and "No PS" column of Table 2 reveals that full parameter sharing enhances returns in 11 out of 18 tasks without compromising safety. We observe significant improvements on tasks involving at least three agents in *MuJoCo* environment [24]: a 38.4-fold increase on the "3x1Hopper" task with three agents, a 2.3-fold increase on the "4x2Ant" task with four agents, and a 3.5-fold increase on the "6x1HalfCheetah" task with six agents.

To verify the role of full parameter sharing in stabilizing training, we visualize the training process on the "3x1Hopper" task with full and no parameter sharing in Fig. 4. Fig. 4 (a) shows a constantly rising return curve with full parameter sharing, contrasting with a persistently low return curve without it. Focusing on the case without full parameter sharing in Fig. 4 (b), the return curve exhibits a fluctuation with a sharp decline in the early training stage, indicating instability. Additionally, the absolute variation coefficient across multiple runs is 0.27 and 0.35 (averaged over all tasks) with full and no parameter sharing, further showing the stabilizing effect of full parameter sharing.

Fig. 4 (c) compares the cumulative cost curves with full and no parameter sharing. With full parameter sharing, MOSDT engages in more aggressive exploration during training, achieving an effective return-cost tradeoff.

To benchmark against partial parameter sharing, we keep the sharing of all parameters except the student action heads $H_S^i$. The result is shown in the "Partial PS" column of Table 2, indicating that performance degradation occurs on 12 out of 18 tasks. Training MOSDT without parameter sharing for any one of the network components causes performance loss on 12 to 14 (out of 18) tasks (cf. Table 8 in Appendix B.4).

In Fig. 3, the comparison between the "MOSDT" and "No PS" columns shows that full parameter sharing substantially reduces the number of training parameters by $47\%$ and the number of execution parameters by $58\%$, contributing to the scalability of MOSDT. Full parameter sharing introduces no additional time consumption in both training and execution (cf. Table 5 in Appendix B.2).

**The effect of cost CBE** To demonstrate the efficacy of CBE, we set cost targets as the given threshold and separately send return features and cumulative cost features into the causal transformer as CDT [7]. The comparison between the "MOSDT" and "w/o CBE" columns in Table 2 reveals that CBE increases returns on 14 out of 18 tasks, covering all base environments including *MuJoCo* [24], *Safety Gym* [25], and *Isaac Gym* [26]. In addition, CBE introduces no additional parameter requirement or computational burden (cf. Appendix B.2). As a plug-and-play method, CBE can be seamlessly integrated into existing safe RL algorithms.

The comparison between the "MOSDT" and "w/o SE" columns in Table 2 shows that the safety signal embedding boosts returns on 12 out of 18 tasks. It enables MOSDT to always access the prior information of correlations between returns and cumulative costs, avoiding MOSDT overlooking either of them (whichever arises later in the input sequence).

According to the training curves (cf. the link in the abstract), CBE has three main advantages on training dynamics:

- Stabilizing training. On most tasks (e.g., "2x3HalfCheetah", "2x3Walker2d", and "3x1Hopper"), CBE results in smoother learning curves, compared with MOSDT without CBE. CBE mitigates volatility and reduces sharp declines. The cost binarization in CBE eliminates the noise in the cost information, thereby stabilizing the training.

- Better training initialization. In most tasks (e.g., "2x4Ant", "4x2Ant", and "Multi-Point1"), CBE yields higher returns with safe cumulative costs at the beginning of training. CBE directly transmits the prior information about the reward-cost correlation to MOSDT by embedding safety signals into returns, enabling agents to learn more efficiently. The cost binarization in CBE eliminates the noise of cost data, improving data efficiency, thereby offering a better training initialization.

- Better safety control. In most tasks (e.g., "2x1Swimmer", "2x4Ant", and "OverFingerMA"), CBE yields cumulative cost learning curves with faster and more stable convergence. For 11 out of 18 tasks, CBE reduces the cumulative costs (cf. the ablation study results in Table 2). CBE provides better safety control by improving the processing of cost information.

**Summary**  PSD is indispensable for safety, while full parameter sharing and CBE deliver the most substantial and the broadest return enhancements, respectively. These three components together enable MOSDT to address the challenges of multi-agent cooperation, offline learning, and safety assurance, achieving SOTA overall performance on MOSDB. Moreover, they reduce parameter counts and time consumption instead of introducing additional overhead.

### 4.3  Analysis of unsafe cases

There are three major causes for the violation of safety constraints:

- High risk in the early stage of training. As shown in the training curves (cf. the link in the abstract), most failure cases show excessive cumulative costs at the initial stages of training. For example, on the "2x3HalfCheetah" task (the cost limit is 25), when $10\%$ of the training (10k steps) is completed, the complete MOSDT only yields a cumulative cost of 13.1, while the MOSDT without PSD yields a cumulative cost of 18.8, and CDT [7] yields a cumulative cost of 62, resulting the violation of safety constraints of the latter two methods (cf. Tables 1 and 2).

- Instability of the cumulative costs in training. In most failure cases, the training curves of cumulative costs show severe fluctuations (e.g., the training curves of BCQ-Lag [8] on the "Multi-Point2" task). This instability indicates that the model often enters unsafe states and cannot maintain safety.

- Lack of communication among agents. Maintaining global safety requires cooperation among agents. Lack of communication among agents could cause them to violate safety constraints. For example, in the "2x3HalfCheetah" task, rollout visualizations of the MOSDT without PSD (i.e., lacking centralized training) revealed that the agent's hind leg persistently accelerated, while its front leg remained stationary to decelerate. This uncoordinated action led to over-speeding and eventual task failure (cf. Table 2).

## 5  Conclusion

This work lays the foundation for MOSRL, providing its first algorithm MOSDT with the first dataset and benchmark MOSDB. Leveraging a novel architecture integrating PSD, full parameter sharing, and CBE, MOSDT achieves a superior tradeoff among returns, safety, and scalability on MOSDB, compared to SOTA single-agent offline safe RL methods (in the CTCE framework).

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

# Appendix

## A    More implementation details

### A.1    Network details of MOSDT

Each linear layer is a single-layer fully connected network. The linear layers for encoding returns and safety signals are 64-dimensional, and other linear layers are 128-dimensional. MOSDT contains 3 causal transformer layers with 128 dimensions and 8 attention heads. Layer Normalization (LN) [43] and Dropout [44] (drop ratio is 0.1) are added before the student causal transformer. We adopt LNs subsequent to both student and teacher causal transformers.

The features before the student causal transformer are arranged in an ordered sequence $\left( f_{RB,1}^i, f_{o,1}^i, f_{a,1}^i, \ldots, f_{RB,M}^i, f_{o,M}^i, f_{a,M}^i \right)$, where $M$ is the training sequence length. The refined feature sequence is divided into 3 subsequences $\left( f_{*,1}^i, \ldots, f_{*,M}^i \right)$ ($* \in \{RB, o, a\}$).

### A.2    More training and evaluation settings

For baseline methods, we adopt the same training and evaluation hyperparameters as DSRL [29], except for the batch size, which is set to 64 in all baseline methods (so as the proposed MOSDT) to reduce training time and yield better performance. For MOSDT, we adopt the same hyperparameters as CDT (one of the baseline methods). The learning rate is $1 \times 10^{-4}$. We use the Adam optimizer [45] with $\beta_1 = 0.9$ and $\beta_2 = 0.999$. The value of Weight Decay is $1 \times 10^{-4}$. We clip gradients to a maximum value of 0.25. We take the first 500 training steps as a warm-up [46].

Following HDT [33], in evaluation, we let CDT [7] and MOSDT perform each execution twice, with the return target of the maximum value in training data and half of the maximum value (the better result is recorded as the final performance). Unlike HDT, we do not set large return targets as they degrade performance.

### A.3    MOSDT algorithm

Algorithm 1 and Algorithm 2 summarize the training and execution processes of MOSDT, respectively.

---

**Algorithm 1** MOSDT training algorithm

---

**Require:** Offline dataset $\mathcal{D}$
1: **for** each training step **do**
2:     Sample trajectory mini-batch $\mathcal{B} \sim \mathcal{D}$. $\mathcal{B}$ contains $\mathcal{T}_M^i = \left( R_1^i, C_1^i, o_1^i, a_1^i, \ldots, R_M^i, C_M^i, o_M^i \right)$ and $a_M^i$, where $M$ is the training sequence length
3:         **for** each agent $i$ **do**
4:             Map $C_t^i$ to $B_t^i$ by Eq. (7)
5:             Generate student features $F_{*,t}^i$ and action estimations $\hat{a}_t^i$ by a student network $\pi_S^i$
6:             Compute the student supervised learning loss $Loss_S^i$ by Eq. (1)
7:         **end for**
8:         Generate a global feature $F_{*,t}$ by Eq. (2)
9:         Generate action estimations $\hat{a}_t^i$ by teacher networks $\pi_T^i$
10:        Compute the teacher supervised learning loss $Loss_T^i$ by Eq. (3)
11:        Compute the PSD loss $Loss_D^i$ by Eq. (4)
12:        Update $\pi_S^i$ (full parameter shared across all agents) and $\pi_T^i$ by minimizing the total loss in Eq. (5)
13: **end for**

---

## B    Further experimental results

The training and evaluation settings are consistent with those in Section 4.1.

---
**Algorithm 2** MOSDT execution algorithm
---
**Require:** Trained agent policies $\pi_S^i$, environment Env)
 1: Get an initial observation: $o_1^i \leftarrow$ Env.reset()
 2: Initialize an input sequence $\mathcal{T}_1^i \leftarrow \left( R_1^i = R^i, B_1^i = 0, o_1^i \right)$, where $R^i$ is predetermined large return targets (task-related).
 3: **for** $t = 1, \ldots, \mathcal{M}$ ($\mathcal{M}$ is the maximum episode length) **do**
 4:     **for** each agent $i$ **do**
 5:         Predict action $\hat{a}_t^i$ from $\mathcal{T}_t^i$ by the trained policy $\pi_S^i$
 6:     **end for**
 7:     Execute the action: $o_{t+1}^i, r_t^i, c_t^i \leftarrow$ Env.step $\left( \hat{a}_t^1, \ldots, \hat{a}_t^N \right)$
 8:     Compute input sequences for the next step:
 9:     $\mathcal{T}_{t+1}^i \leftarrow \left( R_{t+1}^i = R_t^i - r_t^i, B_{t+1}^i = 0, o_{t+1}^i \right)$
10:     **if** Env.terminated() = True **then**
11:         Break
12:     **end if**
13: **end for**
---

## B.1 The performance of BC-Safe

Table 1 presents the performance of the methods trained on the entire MOSDB dataset. The performance of BC-Safe [7] (trained by only safe trajectories) is shown in Table 3. It achieves SOTA return on no task.

Table 3: The performance of BC-Safe [7] on the MOSDB dataset. Results are in the "return $\pm$ sample standard deviation (cumulative cost $\pm$ sample standard deviation)" format. The cost threshold $c$ is 25, and the maximum return of each task is shown in Table 13 in Appendix C.

| Task | BC-Safe [7] |
|---|---|
| **MOS Velocity** | |
| 2x1Swimmer | $5.79 \pm 1.05$ $(2.30 \pm 1.97)$ |
| 2x3HalfCheetah | $2041.64 \pm 13.71$ $(19.60 \pm 2.61)$ |
| 2x3Walker2d | $1562.08 \pm 42.31$ $(1.10 \pm 1.91)$ |
| 2x4Ant | $2406.86 \pm 63.19$ $(0.00 \pm 0.00)$ |
| 3x1Hopper | $28.16 \pm 5.96$ $(0.70 \pm 0.82)$ |
| 4x2Ant | $803.14 \pm 54.20$ $(0.00 \pm 0.00)$ |
| 6x1HalfCheetah | $407.18 \pm 61.72$ $(0.00 \pm 0.00)$ |
| 9\|8Humanoid | $563.67 \pm 40.13$ $(20.37 \pm 0.38)$ |
| **MOS Goal** | |
| Multi-Ant1 | $17.18 \pm 4.58$ $(17.50 \pm 9.54)$ |
| Multi-Ant2 | $1.63 \pm 0.13$ $(7.67 \pm 6.75)$ |
| Multi-Point1 | $5.53 \pm 2.68$ $(10.33 \pm 9.61)$ |
| Multi-Point2 | $0.83 \pm 0.75$ $(12.33 \pm 11.79)$ |
| **MOS Isaac Gym** | |
| CloseDrawerMA | $-5.23 \pm 0.45$ $(1.00 \pm 1.73)$ |
| PickAndPlaceMA | $-5.72 \pm 0.57$ $(3.73 \pm 4.03)$ |
| CatchFingerMA | $0.19 \pm 0.03$ $(0.00 \pm 0.00)$ |
| CatchJointMA | $0.21 \pm 0.01$ $(0.00 \pm 0.00)$ |
| OverFingerMA | $0.43 \pm 0.01$ $(0.00 \pm 0.00)$ |
| OverJointMA | $0.45 \pm 0.02$ $(0.00 \pm 0.00)$ |
| Summary | 0 SOTA (safe) |

## B.2 Detailed efficiency analysis

Table 4 summarizes the parameter counts of CDT [7], MOSDT, and MOSDT variants. Table 5 summarizes their time consumption for training and execution. Compared with CDT [7], MOSDT requires only $65\%$ of the execution parameters and consumes less execution time while achieving better performance. MOSDT needs about twice the training time of CDT. PSD, full parameter sharing, and CBE reduce parameter counts and time consumption rather than introducing additional overhead.

Table 4: The details of parameter counts. Results are in millions (M). "TP": Total parameter count. "EP": Execution parameter count. "Using KD": Using conventional KD instead of PSD. "No PS": No parameter sharing.

| Task | CDT [7] | | MOSDT | | Using KD | | No PS | | w/o CBE | |
|------|------|------|------|------|------|------|------|------|------|------|
| | TP | EP | TP | EP | TP | EP | TP | EP | TP | EP |
| **MOS Velocity** | | | | | | | | | | |
| 2x1Swimmer | 0.7315 | 0.7287 | 0.7771 | 0.3468 | 0.9102 | 0.3468 | 1.1238 | 0.6935 | 0.7774 | 0.3470 |
| 2x3HalfCheetah | 0.7377 | 0.7326 | 0.7801 | 0.3487 | 0.9159 | 0.3487 | 1.1287 | 0.6974 | 0.7803 | 0.3489 |
| 2x3Walker2d | 0.7377 | 0.7326 | 0.7801 | 0.3487 | 0.9159 | 0.3487 | 1.1287 | 0.6974 | 0.7803 | 0.3489 |
| 2x4Ant | 0.7436 | 0.7359 | 0.7822 | 0.3503 | 0.9209 | 0.3503 | 1.1326 | 0.7007 | 0.7825 | 0.3506 |
| 3x1Hopper | 0.7376 | 0.7319 | 0.7944 | 0.3473 | 0.9304 | 0.3473 | 1.4889 | 1.0418 | 0.7946 | 0.3475 |
| 4x2Ant | 0.7606 | 0.7443 | 0.8147 | 0.5481 | 1.3584 | 0.5481 | 2.4590 | 2.1924 | 0.8150 | 0.5484 |
| 6x1HalfCheetah | 0.7634 | 0.7454 | 0.8458 | 0.5467 | 1.3911 | 0.5467 | 3.5793 | 3.2801 | 0.8461 | 0.5469 |
| 9|8Humanoid | 0.9265 | 0.8287 | 0.8314 | 0.5952 | 1.4571 | 0.5952 | 1.4266 | 1.1904 | 0.8317 | 0.5955 |
| **MOS Goal** | | | | | | | | | | |
| Multi-Ant1 | 0.8398 | 0.7853 | 0.8090 | 0.5733 | 1.3916 | 0.5733 | 1.3823 | 1.1467 | 0.8093 | 0.5736 |
| Multi-Ant2 | 0.8398 | 0.7853 | 0.8090 | 0.5733 | 1.3916 | 0.5733 | 1.3823 | 1.1467 | 0.8093 | 0.5736 |
| Multi-Point1 | 0.8063 | 0.7663 | 0.7964 | 0.5638 | 1.3631 | 0.5638 | 1.3603 | 1.1277 | 0.7967 | 0.5641 |
| Multi-Point2 | 0.8063 | 0.7663 | 0.7964 | 0.5638 | 1.3631 | 0.5638 | 1.3603 | 1.1277 | 0.7967 | 0.5641 |
| **MOS Isaac Gym** | | | | | | | | | | |
| CloseDrawerMA | 0.6264 | 0.6146 | 0.6623 | 0.4261 | 1.0749 | 0.4261 | 1.0885 | 0.8523 | 0.6626 | 0.4264 |
| PickAndPlaceMA | 0.6290 | 0.6158 | 0.6630 | 0.4268 | 1.0769 | 0.4268 | 1.0897 | 0.8535 | 0.6632 | 0.4270 |
| CatchFingerMA | 0.7334 | 0.6756 | 0.6997 | 0.4547 | 1.1630 | 0.4547 | 1.1544 | 0.9094 | 0.6999 | 0.4550 |
| CatchJointMA | 0.7334 | 0.6756 | 0.6997 | 0.4547 | 1.1630 | 0.4547 | 1.1544 | 0.9094 | 0.6999 | 0.4550 |
| OverFingerMA | 0.7226 | 0.6679 | 0.6927 | 0.4509 | 1.1514 | 0.4509 | 1.1436 | 0.9017 | 0.6930 | 0.4511 |
| OverJointMA | 0.7226 | 0.6679 | 0.6927 | 0.4509 | 1.1514 | 0.4509 | 1.1436 | 0.9017 | 0.6930 | 0.4511 |
| Average | 0.7554 | 0.7223 | 0.7626 | 0.4650 | 1.1717 | 0.4650 | 1.4293 | 1.1317 | 0.7629 | 0.4653 |

### B.3 Detailed ablation study for PSD

Table 6 shows the results of detailed ablation experiments for PSD (the sample standard deviation of each result is presented in Table 7).

The "w/o ASD" column and the "w/o FSD" column in Table 6 show that removing the action/feature distillation (the two terms of Eq. (4) reduces returns on 13/12 (out of 18) tasks.

The "Using SRSD" column in Table 6 shows that the structural relation distillation in MADTKD [5] is not well compatible with the MOSDB dataset or the MOSDT architecture.

### B.4 Detailed ablation study for partial parameter sharing

Table 8 shows the results of partial parameter sharing (the sample standard deviations are presented in Table 9). Partial parameter sharing leads to performance losses on 12 to 14 (out of 18) tasks.

### B.5 Ablation study for network designs

Table 10 presents the results of detaching the agent features $F_{*,t}^i$ from the computational graph of Eq. (2), showing performance degradation on 10 out of 18 tasks.

### B.6 Standard deviation results

Table 11 shows the sample standard deviation of each value in Table 1. Table 12 shows the sample standard deviation of each value in Table 2.

## C More details of the MOSDB dataset

Table 13 shows the details of the MOSDB dataset. Fig. 5 presents the data distribution over the reward-cost plane of each task in MOSDB.

Table 5: The details of time consumption. Results are in milliseconds (ms). "TT": The time consumption of one training step. "ET": The time consumption of an inference for one agent. "Using KD": Using conventional KD instead of PSD. "No PS": No parameter sharing. The results in "Using KD-ET" are identical to those in "MOSDT-ET", because using KD has no effect during execution.

| Task | CDT [7] | | MOSDT | | Using KD | | No PS | | w/o CBE | |
|---|---|---|---|---|---|---|---|---|---|---|
| | TP | EP | TP | EP | TP | EP | TP | EP | TP | EP |
| **MOS Velocity** | | | | | | | | | | |
| 2x1Swimmer | 6.65 | 0.76 | 11.19 | 0.54 | 15.27 | 0.54 | 11.68 | 0.56 | 10.92 | 0.55 |
| 2x3HalfCheetah | 6.06 | 0.75 | 11.30 | 0.54 | 14.15 | 0.54 | 11.23 | 0.55 | 10.97 | 0.55 |
| 2x3Walker2d | 6.21 | 0.92 | 10.97 | 0.58 | 14.35 | 0.58 | 11.77 | 0.56 | 11.02 | 0.54 |
| 2x4Ant | 6.28 | 0.76 | 11.13 | 0.56 | 14.42 | 0.56 | 11.61 | 0.58 | 11.14 | 0.56 |
| 3x1Hopper | 6.19 | 0.92 | 13.84 | 0.59 | 16.84 | 0.59 | 14.88 | 0.66 | 14.71 | 0.57 |
| 4x2Ant | 6.24 | 0.77 | 18.75 | 0.71 | 22.13 | 0.71 | 20.51 | 0.73 | 19.42 | 0.68 |
| 6x1HalfCheetah | 6.32 | 0.76 | 26.44 | 0.71 | 29.63 | 0.71 | 30.13 | 0.74 | 26.33 | 0.70 |
| 9\|8Humanoid | 7.21 | 0.88 | 12.84 | 0.85 | 17.42 | 0.85 | 12.45 | 0.78 | 13.00 | 0.76 |
| **MOS Goal** | | | | | | | | | | |
| Multi-Ant1 | 6.85 | 0.79 | 11.76 | 0.74 | 16.70 | 0.74 | 12.17 | 0.74 | 12.22 | 0.75 |
| Multi-Ant2 | 6.59 | 0.78 | 12.30 | 0.72 | 16.59 | 0.72 | 12.08 | 0.75 | 12.35 | 0.76 |
| Multi-Point1 | 6.65 | 0.78 | 11.95 | 0.73 | 16.40 | 0.73 | 12.22 | 0.76 | 11.97 | 0.73 |
| Multi-Point2 | 6.48 | 0.79 | 11.95 | 0.75 | 16.32 | 0.75 | 12.65 | 0.78 | 12.07 | 0.75 |
| **MOS Isaac Gym** | | | | | | | | | | |
| CloseDrawerMA | 6.63 | 1.02 | 11.61 | 1.08 | 16.46 | 1.08 | 12.05 | 1.10 | 11.53 | 1.41 |
| PickAndPlaceMA | 6.62 | 1.03 | 11.57 | 1.07 | 16.35 | 1.07 | 11.82 | 1.09 | 11.65 | 1.07 |
| CatchFingerMA | 6.68 | 0.94 | 12.11 | 1.06 | 16.72 | 1.06 | 12.53 | 1.11 | 12.01 | 1.05 |
| CatchJointMA | 6.81 | 0.93 | 12.81 | 1.06 | 17.05 | 1.06 | 12.79 | 1.12 | 12.49 | 1.06 |
| OverFingerMA | 6.80 | 0.93 | 12.21 | 1.08 | 16.62 | 1.08 | 12.72 | 1.08 | 11.95 | 1.05 |
| OverJointMA | 6.82 | 0.93 | 12.53 | 1.08 | 16.88 | 1.08 | 13.33 | 1.14 | 11.95 | 1.08 |
| Average | 6.56 | 0.86 | 13.18 | 0.80 | 17.24 | 0.80 | 13.81 | 0.82 | 13.21 | 0.81 |

# D Limitations and future work

MOSRL may be used for negative applications such as drone surveillance. Upon closer inspection, no data at high risk of such misuse is observed in MOSDB. In addition, the cost binarization in CBE may constrain the adaptability of models to dynamic cost thresholds. This drawback may not be an issue in many applications with fixed cost thresholds [10, 47]. Future efforts will focus on expanding the MOSDB dataset.

Table 6: The results of detailed ablation experiments for PSD. Results are in the "return (cumulative cost)" format. The cost threshold $c$ is 25. The maximum return of each task is shown in Table 13 in Appendix C. "⇓": Policies with lower returns than full MOSDT or unsafe policies. Red: Unsafe policies. "w/o ASD": Without action self-distillation. "w/o FSD": Without feature self-distillation. "Using SRSD": Adding a structural relation distillation [5] term in Eq. (4). Due to space constraints, the sample standard deviation across multiple runs of each experiment is shown in Table 7.

| Task | MOSDT | w/o ASD | w/o FSD | Using SRSD |
|---|---|---|---|---|
| **MOS Velocity** | | | | |
| 2x1Swimmer | 11.64 (20.33) | ⇓ 9.48 (10.20) | ⇓ 11.43 (10.67) | ⇓ 11.02 (12.53) |
| 2x3HalfCheetah | 2052.64 (22.27) | 2074.78 (24.17) | ⇓ 2084.67 (35.43) | ⇓ 2050.06 (24.00) |
| 2x3Walker2d | 1584.87 (3.83) | ⇓ 1583.14 (3.07) | ⇓ 1529.36 (0.10) | ⇓ 1562.50 (4.57) |
| 2x4Ant | 2054.88 (0.87) | ⇓ 1971.57 (1.97) | ⇓ 2045.89 (5.60) | ⇓ 1975.55 (0.90) |
| 3x1Hopper | 1122.23 (4.00) | 1236.94 (7.10) | 1199.86 (11.43) | ⇓ 1067.13 (8.53) |
| 4x2Ant | 2083.85 (3.53) | ⇓ 2080.61 (2.93) | 2191.79 (3.50) | 2093.81 (0.57) |
| 6x1HalfCheetah | 1853.64 (21.97) | ⇓ 1835.71 (23.97) | ⇓ 1808.95 (25.73) | 1862.54 (22.80) |
| 9\|8Humanoid | 444.71 (22.80) | 477.64 (21.73) | 510.74 (21.47) | 460.61 (22.60) |
| **MOS Goal** | | | | |
| Multi-Ant1 | 38.38 (14.50) | ⇓ 30.11 (11.33) | 44.95 (14.33) | ⇓ 35.30 (14.17) |
| Multi-Ant2 | 2.96 (7.50) | ⇓ 2.85 (15.33) | ⇓ 3.82 (13.50) | ⇓ 1.55 (8.33) |
| Multi-Point1 | 9.65 (12.67) | ⇓ 5.61 (23.00) | 11.33 (20.83) | 11.88 (21.50) |
| Multi-Point2 | -1.08 (21.00) | ⇓ -1.79 (15.00) | ⇓ -4.35 (8.33) | 0.85 (12.67) |
| **MOS Isaac Gym** | | | | |
| CloseDrawerMA | -3.45 (0.00) | ⇓ -3.45 (0.00) | ⇓ -3.73 (0.00) | -3.44 (0.67) |
| PickAndPlaceMA | -2.67 (0.00) | -2.54 (0.00) | -2.07 (0.00) | ⇓ -2.87 (0.00) |
| CatchFingerMA | 0.25 (6.33) | ⇓ 0.21 (4.47) | ⇓ 0.17 (5.57) | 0.27 (3.50) |
| CatchJointMA | 0.31 (0.00) | ⇓ 0.31 (0.00) | ⇓ 0.23 (0.00) | ⇓ 0.23 (0.00) |
| OverFingerMA | 0.52 (0.00) | ⇓ 0.46 (0.00) | ⇓ 0.43 (1.97) | ⇓ 0.44 (0.00) |
| OverJointMA | 0.47 (1.03) | 0.48 (0.00) | ⇓ 0.44 (1.60) | ⇓ 0.46 (0.00) |
| Summary | (safe) | 13 ⇓ (safe) | 12 ⇓ (safe) | 11 ⇓ (safe) |

Table 7: The results of detailed ablation experiments for PSD (sample standard deviation). Results are in the "return sample standard deviation (cumulative cost sample standard deviation)" format. "w/o ASD": Without action distillation. "w/o FSD": Without feature distillation. "Using SRSD": Adding a structural relation distillation [5] term in Eq. (4)

| Task | MOSDT | w/o ASD | w/o FSD | Using SRSD |
|---|---|---|---|---|
| **MOS Velocity** | | | | |
| 2x1Swimmer | 1.41 (2.61) | 1.43 (3.37) | 1.05 (4.92) | 2.43 (5.07) |
| 2x3HalfCheetah | 23.79 (1.88) | 4.53 (0.57) | 37.07 (6.09) | 10.40 (0.89) |
| 2x3Walker2d | 23.95 (5.43) | 52.60 (2.80) | 28.64 (0.17) | 52.20 (7.39) |
| 2x4Ant | 73.62 (1.33) | 10.77 (2.18) | 32.35 (2.56) | 48.81 (0.90) |
| 3x1Hopper | 131.42 (4.65) | 40.69 (8.02) | 4.59 (6.48) | 167.69 (9.86) |
| 4x2Ant | 64.59 (3.39) | 41.88 (0.76) | 21.65 (1.21) | 48.40 (0.15) |
| 6x1HalfCheetah | 42.95 (1.15) | 4.24 (1.21) | 26.32 (1.26) | 19.34 (1.54) |
| 9\|8Humanoid | 8.88 (0.10) | 31.93 (0.35) | 11.80 (0.71) | 11.29 (0.92) |
| **MOS Goal** | | | | |
| Multi-Ant1 | 0.84 (12.56) | 11.11 (8.96) | 3.40 (10.13) | 8.19 (11.00) |
| Multi-Ant2 | 1.56 (8.35) | 0.85 (5.11) | 1.39 (6.54) | 0.57 (14.43) |
| Multi-Point1 | 2.52 (1.53) | 0.71 (1.50) | 2.94 (3.51) | 4.00 (2.18) |
| Multi-Point2 | 2.89 (4.09) | 2.40 (6.06) | 3.95 (4.54) | 3.23 (8.39) |
| **MOS Isaac Gym** | | | | |
| CloseDrawerMA | 0.38 (0.00) | 0.38 (0.00) | 0.03 (0.00) | 0.58 (1.15) |
| PickAndPlaceMA | 0.52 (0.00) | 1.15 (0.00) | 0.27 (0.00) | 1.07 (0.00) |
| CatchFingerMA | 0.03 (0.15) | 0.04 (2.60) | 0.01 (0.85) | 0.11 (0.75) |
| CatchJointMA | 0.09 (0.00) | 0.10 (0.00) | 0.03 (0.00) | 0.01 (0.00) |
| OverFingerMA | 0.10 (0.00) | 0.04 (0.00) | 0.01 (2.89) | 0.01 (0.00) |
| OverJointMA | 0.03 (1.05) | 0.05 (0.00) | 0.03 (0.72) | 0.04 (0.00) |

Table 8: The results of partial parameter sharing. Results are in the "return (cumulative cost)" format. The cost threshold $c$ is 25. The maximum return of each task is shown in Table 13 in Appendix C. "⇓": Policies with lower returns than full MOSDT or unsafe policies. "w/o PSH": Without parameter sharing for student action heads. "w/o PSFP": Without parameter sharing for student feature projector. "w/o PST": Without parameter sharing for student causal transformers. "w/o PSE": Without parameter sharing for encoders. Due to space constraints, the sample standard deviation across multiple runs of each experiment is shown in Table 9.

| Task | MOSDT | Partial parameter sharing | | | |
| --- | --- | --- | --- | --- | --- |
| | | w/o PSH | w/o PSFP | w/o PST | w/o PSE |
| **MOS Velocity** | | | | | |
| 2x1Swimmer | 11.64 (20.33) | ⇓ 10.97 (15.33) | ⇓ 9.85 (18.10) | ⇓ 10.85 (15.67) | ⇓ 8.37 (3.83) |
| 2x3HalfCheetah | 2052.64 (22.27) | ⇓ 2050.14 (23.00) | ⇓ 2050.25 (23.13) | ⇓ 2049.17 (22.27) | ⇓ 2041.74 (23.87) |
| 2x3Walker2d | 1584.87 (3.83) | 1585.52 (2.07) | ⇓ 1558.54 (2.93) | ⇓ 1578.55 (1.43) | ⇓ 1557.33 (2.80) |
| 2x4Ant | 2054.88 (0.87) | ⇓ 2049.88 (1.73) | ⇓ 1988.90 (0.00) | ⇓ 1983.47 (2.30) | ⇓ 2015.30 (0.33) |
| 3x1Hopper | 1122.23 (4.00) | ⇓ 110.93 (1.33) | ⇓ 47.56 (2.53) | ⇓ 33.54 (0.00) | ⇓ 34.09 (0.03) |
| 4x2Ant | 2083.85 (3.53) | ⇓ 962.59 (0.00) | ⇓ 934.06 (0.00) | ⇓ 910.53 (0.00) | ⇓ 934.31 (0.00) |
| 6x1HalfCheetah | 1853.64 (21.97) | ⇓ 1169.26 (0.10) | ⇓ 407.84 (0.00) | ⇓ 330.70 (0.00) | ⇓ 436.95 (0.00) |
| 9\|8Humanoid | 444.71 (22.80) | 474.87 (21.73) | 489.29 (22.50) | 508.10 (20.77) | 459.40 (21.97) |
| **MOS Goal** | | | | | |
| Multi-Ant1 | 38.38 (14.50) | ⇓ 36.87 (10.50) | ⇓ 36.22 (20.00) | ⇓ 34.36 (17.50) | ⇓ 33.96 (14.00) |
| Multi-Ant2 | 2.96 (7.50) | 4.60 (11.67) | 3.20 (7.33) | 4.29 (19.67) | ⇓ 2.60 (17.67) |
| Multi-Point1 | 9.65 (12.67) | 10.42 (14.83) | 9.70 (14.67) | ⇓ 8.18 (20.17) | 12.41 (19.83) |
| Multi-Point2 | -1.08 (21.00) | 0.33 (7.83) | -0.82 (14.17) | 3.83 (18.17) | ⇓ -3.67 (11.00) |
| **MOS Isaac Gym** | | | | | |
| CloseDrawerMA | -3.45 (0.00) | ⇓ -3.46 (0.00) | ⇓ -3.68 (0.20) | -3.30 (0.67) | -3.28 (0.67) |
| PickAndPlaceMA | -2.67 (0.00) | -2.24 (0.00) | ⇓ -2.85 (0.00) | ⇓ -2.98 (0.00) | -2.24 (1.83) |
| CatchFingerMA | 0.25 (6.33) | ⇓ 0.23 (5.33) | ⇓ 0.20 (3.10) | ⇓ 0.20 (7.03) | ⇓ 0.22 (7.23) |
| CatchJointMA | 0.31 (0.00) | ⇓ 0.27 (0.00) | 0.32 (0.00) | 0.36 (0.00) | ⇓ 0.25 (0.00) |
| OverFingerMA | 0.52 (0.00) | ⇓ 0.48 (0.00) | ⇓ 0.49 (0.00) | ⇓ 0.46 (0.00) | ⇓ 0.49 (0.00) |
| OverJointMA | 0.47 (1.03) | ⇓ 0.46 (0.00) | ⇓ 0.46 (0.00) | ⇓ 0.46 (0.00) | ⇓ 0.46 (0.00) |
| Summary | (safe) | 12 ⇓ (safe) | 13 ⇓ (safe) | 13 ⇓ (safe) | 14 ⇓ (safe) |

Table 9: The results of partial parameter sharing (sample standard deviation). Results are in the "return sample standard deviation (cumulative cost sample standard deviation)" format. "w/o PSH": Without parameter sharing for student action heads. "w/o PSFP": Without parameter sharing for student feature projector. "w/o PST": Without parameter sharing for student causal transformers. "w/o PSE": Without parameter sharing for encoders.

| Task | MOSDT | Partial parameter sharing | | | |
| --- | --- | --- | --- | --- | --- |
| | | w/o PSH | w/o PSFP | w/o PST | w/o PSE |
| **MOS Velocity** | | | | | |
| 2x1Swimmer | 1.41 (2.61) | 2.51 (3.19) | 2.41 (4.12) | 3.46 (3.61) | 1.47 (2.40) |
| 2x3HalfCheetah | 23.79 (1.88) | 4.77 (0.89) | 19.05 (1.66) | 0.84 (0.90) | 14.62 (0.21) |
| 2x3Walker2d | 23.95 (5.43) | 59.87 (1.37) | 66.45 (2.11) | 12.86 (1.45) | 2.44 (4.85) |
| 2x4Ant | 73.62 (1.33) | 147.35 (1.70) | 14.82 (0.00) | 51.64 (2.89) | 58.57 (0.35) |
| 3x1Hopper | 131.42 (4.65) | 125.36 (2.31) | 10.97 (4.39) | 6.79 (0.00) | 5.46 (0.06) |
| 4x2Ant | 64.59 (3.39) | 9.76 (0.00) | 16.76 (0.00) | 4.02 (0.00) | 22.29 (0.00) |
| 6x1HalfCheetah | 42.95 (1.15) | 115.20 (0.17) | 67.88 (0.00) | 24.06 (0.00) | 67.84 (0.00) |
| 9\|8Humanoid | 8.88 (0.10) | 10.98 (0.75) | 29.37 (0.36) | 60.67 (1.64) | 24.41 (0.35) |
| **MOS Goal** | | | | | |
| Multi-Ant1 | 0.84 (12.56) | 9.15 (9.73) | 4.86 (3.61) | 4.25 (3.28) | 3.81 (3.12) |
| Multi-Ant2 | 1.56 (8.35) | 2.54 (7.52) | 1.11 (6.83) | 2.29 (6.21) | 1.83 (5.39) |
| Multi-Point1 | 2.52 (1.53) | 3.23 (8.22) | 6.89 (6.11) | 2.33 (3.18) | 2.69 (2.02) |
| Multi-Point2 | 2.89 (4.09) | 2.20 (7.75) | 1.86 (10.75) | 4.30 (4.48) | 5.26 (10.76) |
| **MOS Isaac Gym** | | | | | |
| CloseDrawerMA | 0.38 (0.00) | 0.32 (0.00) | 0.44 (0.35) | 0.43 (1.15) | 0.33 (1.15) |
| PickAndPlaceMA | 0.52 (0.00) | 0.41 (0.00) | 0.75 (0.00) | 0.25 (0.00) | 0.47 (3.18) |
| CatchFingerMA | 0.03 (0.15) | 0.08 (2.31) | 0.04 (2.04) | 0.02 (0.95) | 0.06 (0.70) |
| CatchJointMA | 0.09 (0.00) | 0.06 (0.00) | 0.09 (0.00) | 0.09 (0.00) | 0.09 (0.00) |
| OverFingerMA | 0.10 (0.00) | 0.07 (0.00) | 0.03 (0.00) | 0.02 (0.00) | 0.05 (0.00) |
| OverJointMA | 0.03 (1.05) | 0.03 (0.00) | 0.03 (0.00) | 0.01 (0.00) | 0.03 (0.00) |

Table 10: The results of detaching agent features. Results are in the "return $\pm$ sample standard deviation (cumulative cost $\pm$ sample standard deviation)" format. The cost threshold $c$ is 25. The maximum return of each task is shown in Table 13 in Appendix C. "$\Downarrow$": Policies with lower returns than full MOSDT or unsafe policies. "Detaching": Detaching agent features $F_{*,t}^i$ from the computational graph of Eq. (2).

| Task | MOSDT | Detaching |
|---|---|---|
| **MOS Velocity** | | |
| 2x1Swimmer | $11.64 \pm 20.33 \ (1.41 \pm 2.61)$ | $\Downarrow 8.60 \pm 2.13 \ (12.70 \pm 9.40)$ |
| 2x3HalfCheetah | $2052.64 \pm 22.27 \ (23.79 \pm 1.88)$ | $2054.66 \pm 7.79 \ (23.07 \pm 1.53)$ |
| 2x3Walker2d | $1584.87 \pm 3.83 \ (23.95 \pm 5.43)$ | $\Downarrow 1557.64 \pm 13.47 \ (2.37 \pm 2.30)$ |
| 2x4Ant | $2054.88 \pm 0.87 \ (73.62 \pm 1.33)$ | $\Downarrow 2006.04 \pm 46.42 \ (0.63 \pm 0.93)$ |
| 3x1Hopper | $1122.23 \pm 4.00 \ (131.42 \pm 4.65)$ | $1203.32 \pm 93.29 \ (13.13 \pm 3.86)$ |
| 4x2Ant | $2083.85 \pm 3.53 \ (64.59 \pm 3.39)$ | $2114.28 \pm 31.37 \ (6.13 \pm 8.11)$ |
| 6x1HalfCheetah | $1853.64 \pm 21.97 \ (42.95 \pm 1.15)$ | $1869.79 \pm 29.71 \ (23.30 \pm 0.62)$ |
| 9\|8Humanoid | $444.71 \pm 22.80 \ (8.88 \pm 0.10)$ | $460.98 \pm 2.91 \ (21.50 \pm 0.95)$ |
| **MOS Goal** | | |
| Multi-Ant1 | $38.38 \pm 14.50 \ (0.84 \pm 12.56)$ | $\Downarrow 28.44 \pm 12.25 \ (15.50 \pm 5.50)$ |
| Multi-Ant2 | $2.96 \pm 7.50 \ (1.56 \pm 8.35)$ | $\Downarrow 2.01 \pm 0.90 \ (16.67 \pm 4.93)$ |
| Multi-Point1 | $9.65 \pm 12.67 \ (2.52 \pm 1.53)$ | $\Downarrow 9.16 \pm 1.89 \ (17.50 \pm 6.76)$ |
| Multi-Point2 | $-1.08 \pm 21.00 \ (2.89 \pm 4.09)$ | $-0.02 \pm 1.16 \ (11.83 \pm 10.75)$ |
| **MOS Isaac Gym** | | |
| CloseDrawerMA | $-3.45 \pm 0.00 \ (0.38 \pm 0.00)$ | $-3.25 \pm 0.32 \ (0.67 \pm 1.15)$ |
| PickAndPlaceMA | $-2.67 \pm 0.00 \ (0.52 \pm 0.00)$ | $-2.43 \pm 0.57 \ (2.67 \pm 4.62)$ |
| CatchFingerMA | $0.25 \pm 6.33 \ (0.03 \pm 0.15)$ | $\Downarrow 0.21 \pm 0.01 \ (4.03 \pm 2.61)$ |
| CatchJointMA | $0.31 \pm 0.00 \ (0.09 \pm 0.00)$ | $\Downarrow 0.30 \pm 0.08 \ (0.00 \pm 0.00)$ |
| OverFingerMA | $0.52 \pm 0.00 \ (0.10 \pm 0.00)$ | $\Downarrow 0.48 \pm 0.04 \ (0.00 \ 0.00)$ |
| OverJointMA | $0.47 \pm 1.03 \ (0.03 \pm 1.05)$ | $\Downarrow 0.47 \pm 0.04 \ (1.60 \pm 2.77)$ |
| Summary | (safe) | $10 \Downarrow$ (safe) |

Table 11: MOSDB benchmark and the performance of MOSDT (sample standard deviation). Results are in the "return sample standard deviation (cumulative cost sample standard deviation)" format.

| Task | BC [7] | BCQ-Lag [8] | BEAR-Lag [8] | CDT [7] | COptiDICE [41] | CPQ [8] | MOSDT (ours) |
|---|---|---|---|---|---|---|---|
| **MOS Velocity** | | | | | | | |
| 2x1Swimmer | 1.03 (2.57) | 0.70 (4.17) | 3.64 (8.04) | 1.65 (6.35) | 0.77 (2.26) | 10.47 (11.48) | 1.41 (2.61) |
| 2x3HalfCheetah | 6.54 (0.97) | 38.19 (12.88) | 67.62 (26.85) | 19.86 (3.29) | 15.20 (1.45) | 607.92 (9.27) | 23.79 (1.88) |
| 2x3Walker2d | 43.72 (0.00) | 87.21 (11.53) | 47.46 (6.03) | 65.25 (3.18) | 12.95 (0.00) | 201.40 (6.30) | 23.95 (5.43) |
| 2x4Ant | 124.08 (0.00) | 8.48 (11.72) | 181.20 (1.56) | 25.58 (2.12) | 48.48 (1.14) | 1156.58 (0.87) | 73.62 (1.33) |
| 3x1Hopper | 4.63 (0.52) | 23.33 (3.09) | 16.17 (3.23) | 2.88 (0.00) | 48.44 (10.25) | 98.51 (1.85) | 131.42 (4.65) |
| 4x2Ant | 19.22 (0.00) | 58.78 (0.00) | 41.27 (0.00) | 22.55 (0.00) | 6.10 (0.00) | 356.92 (0.84) | 64.59 (3.39) |
| 6x1HalfCheetah | 35.43 (0.00) | 87.69 (0.06) | 49.45 (0.00) | 20.85 (0.00) | 38.27 (0.00) | 122.73 (1.50) | 42.95 (1.15) |
| 9\|8Humanoid | 19.84 (1.30) | 24.39 (1.30) | 28.77 (1.96) | 35.03 (0.47) | 21.44 (1.17) | 38.79 (7.79) | 8.88 (0.10) |
| **MOS Goal** | | | | | | | |
| Multi-Ant1 | 7.24 (2.78) | 0.16 (0.96) | 5.80 (12.76) | 10.04 (9.57) | 10.39 (6.73) | 0.34 (0.00) | 0.84 (12.56) |
| Multi-Ant2 | 0.98 (7.09) | 0.16 (1.28) | 1.13 (12.27) | 2.35 (4.09) | 1.24 (7.29) | 0.74 (0.00) | 1.56 (8.35) |
| Multi-Point1 | 4.22 (6.83) | 2.64 (4.75) | 3.84 (5.20) | 0.84 (12.41) | 1.22 (6.83) | 2.21 (13.28) | 2.52 (1.53) |
| Multi-Point2 | 0.78 (7.51) | 4.51 (18.30) | 3.02 (6.24) | 2.23 (5.13) | 0.86 (4.65) | 0.78 (9.50) | 2.89 (4.09) |
| **MOS Isaac Gym** | | | | | | | |
| CloseDrawerMA | 0.45 (1.73) | 0.26 (0.87) | 1.53 (4.07) | 0.43 (0.00) | 0.15 (0.00) | 0.35 (0.00) | 0.38 (0.00) |
| PickAndPlaceMA | 0.57 (4.03) | 0.72 (0.46) | 0.43 (0.00) | 0.54 (0.00) | 1.75 (4.62) | 1.52 (4.27) | 0.52 (0.00) |
| CatchFingerMA | 0.03 (0.00) | 0.01 (1.28) | 0.02 (0.00) | 0.02 (0.10) | 0.05 (0.00) | 0.03 (1.50) | 0.03 (0.15) |
| CatchJointMA | 0.01 (0.00) | 0.04 (0.00) | 0.03 (0.00) | 0.01 (0.70) | 0.06 (0.00) | 0.08 (0.55) | 0.09 (0.00) |
| OverFingerMA | 0.01 (0.00) | 0.02 (0.12) | 0.02 (0.00) | 0.05 (0.00) | 0.03 (0.00) | 0.00 (0.00) | 0.10 (0.00) |
| OverJointMA | 0.02 (0.00) | 0.01 (0.00) | 0.03 (0.00) | 0.01 (1.55) | 0.02 (0.00) | 0.01 (0.00) | 0.03 (1.05) |

Table 12: Ablation study results (sample standard deviation). Results are in the "return sample standard deviation (cumulative cost sample standard deviation)" format. The cost threshold $c$ is 25. The maximum return of each task is shown in Table 13 in Appendix C. "Partial PS": Without parameter sharing for student action heads. "No PS": No parameter sharing. "w/o SE": Without safety signal embedding.

| Task | MOSDT | w/o PSD | Partial PS | No PS | w/o SE | w/o CBE |
|---|---|---|---|---|---|---|
| **MOS Velocity** | | | | | | |
| 2x1Swimmer | 1.41 (2.61) | 1.49 (4.16) | 2.51 (3.19) | 1.89 (2.66) | 1.57 (7.72) | 2.39 (1.41) |
| 2x3HalfCheetah | 23.79 (1.88) | 32.45 (5.24) | 4.77 (0.89) | 21.86 (2.70) | 3.41 (3.86) | 47.33 (5.58) |
| 2x3Walker2d | 23.95 (5.43) | 73.25 (5.52) | 59.87 (1.37) | 89.35 (1.88) | 54.69 (5.08) | 104.87 (4.76) |
| 2x4Ant | 73.62 (1.33) | 20.82 (2.02) | 147.35 (1.70) | 88.48 (10.75) | 34.64 (8.67) | 17.59 (2.46) |
| 3x1Hopper | 131.42 (4.65) | 16.26 (10.93) | 125.36 (2.31) | 2.80 (0.00) | 171.57 (5.65) | 59.61 (7.27) |
| 4x2Ant | 64.59 (3.39) | 22.42 (1.42) | 9.76 (0.00) | 1.83 (0.00) | 48.07 (2.56) | 58.66 (7.48) |
| 6x1HalfCheetah | 42.95 (1.15) | 51.38 (1.80) | 115.20 (0.17) | 82.59 (0.00) | 13.82 (2.42) | 4.23 (0.74) |
| 9\|8Humanoid | 8.88 (0.10) | 18.53 (1.01) | 10.98 (0.75) | 51.44 (0.36) | 24.66 (0.55) | 9.89 (0.15) |
| **MOS Goal** | | | | | | |
| Multi-Ant1 | 0.84 (12.56) | 11.72 (2.25) | 9.15 (9.73) | 11.09 (11.93) | 8.38 (4.01) | 9.81 (4.92) |
| Multi-Ant2 | 1.56 (8.35) | 1.61 (10.32) | 2.54 (7.52) | 0.49 (5.03) | 0.50 (0.90) | 0.33 (4.54) |
| Multi-Point1 | 2.52 (1.53) | 3.02 (5.06) | 3.23 (8.22) | 3.30 (4.86) | 1.46 (8.69) | 0.51 (11.82) |
| Multi-Point2 | 2.89 (4.09) | 1.73 (10.33) | 2.20 (7.75) | 5.12 (3.69) | 0.71 (9.96) | 1.44 (7.77) |
| **MOS Isaac Gym** | | | | | | |
| CloseDrawerMA | 0.38 (0.00) | 0.06 (0.00) | 0.32 (0.00) | 0.48 (0.00) | 0.20 (0.00) | 0.14 (0.00) |
| PickAndPlaceMA | 0.52 (0.00) | 0.26 (0.00) | 0.41 (0.00) | 0.25 (0.00) | 0.69 (4.62) | 0.44 (0.00) |
| CatchFingerMA | 0.03 (0.15) | 0.03 (1.82) | 0.08 (2.31) | 0.02 (3.94) | 0.03 (3.07) | 0.02 (0.53) |
| CatchJointMA | 0.09 (0.00) | 0.05 (0.00) | 0.06 (0.00) | 0.01 (0.00) | 0.06 (0.00) | 0.02 (0.00) |
| OverFingerMA | 0.10 (0.00) | 0.01 (0.92) | 0.07 (0.00) | 0.01 (0.00) | 0.02 (0.00) | 0.15 (3.50) |
| OverJointMA | 0.03 (1.05) | 0.02 (0.44) | 0.03 (0.00) | 0.02 (0.00) | 0.00 (0.00) | 0.09 (3.91) |

Table 13: The details of the MOSDB dataset. "Full name": Full name in Safety Gymnasium [28]. "Observation dimension": The sum of the observation dimensions of all agents. "Action dimension": The sum of action dimensions of all agents. "Credit assignment": "✓" means the environment yields respective returns and costs for each agent. "✗" means the environment yields system returns and costs. "Maximum return": The maximum return in the MOSDB dataset.

| Task | Full name | Number of agents | Observation dimension | Action dimension | Credit assignment | Maximum return |
|---|---|---|---|---|---|---|
| **MOS Velocity** | | | | | | |
| 2x1Swimmer | Safety2x1SwimmerVelocity-v0 | 2 | 20 | 2 | ✗ | 12.99 |
| 2x3HalfCheetah | Safety2x3HalfCheetahVelocity-v0 | 2 | 38 | 6 | ✗ | 1672.97 |
| 2x3Walker2d | Safety2x3Walker2dVelocity-v0 | 2 | 38 | 6 | ✗ | 2031.20 |
| 2x4Ant | Safety2x4AntVelocity-v0 | 2 | 58 | 8 | ✗ | 1910.11 |
| 3x1Hopper | Safety3x1HopperVelocity-v0 | 3 | 54 | 3 | ✗ | 1326.51 |
| 4x2Ant | Safety4x2AntVelocity-v0 | 4 | 124 | 8 | ✗ | 1882.12 |
| 6x1HalfCheetah | Safety6x1HalfCheetahVelocity-v0 | 6 | 138 | 6 | ✗ | 1626.53 |
| 9\|8Humanoid | Safety9\|8HumanoidVelocity-v0 | 2 | 756 | 17 | ✗ | 1438.35 |
| **MOS Goal** | | | | | | |
| Multi-Ant1 | SafetyAntMultiGoal1-v0 | 2 | 420 | 16 | ✓ | 36.76 |
| Multi-Ant2 | SafetyAntMultiGoal2-v0 | 2 | 420 | 16 | ✓ | 8.47 |
| Multi-Point1 | SafetyPointMultiGoal1-v0 | 2 | 308 | 4 | ✓ | 17.99 |
| Multi-Point2 | SafetyPointMultiGoal2-v0 | 2 | 308 | 4 | ✓ | 14.49 |
| **MOS Isaac Gym** | | | | | | |
| CloseDrawerMA | FreightFrankaCloseDrawer | 2 | 90 | 12 | ✓ | 10.30 |
| PickAndPlaceMA | FreightFrankaPickAndPlace | 2 | 100 | 12 | ✓ | -0.83 |
| CatchFingerMA | ShadowHandCatchOver2Underarm_Safe_finger | 2 | 446 | 52 | ✓ | 7.19 |
| CatchJointMA | ShadowHandCatchOver2Underarm_Safe_joint | 2 | 446 | 52 | ✓ | 6.21 |
| OverFingerMA | ShadowHandOver_Safe_finger | 2 | 422 | 40 | ✓ | 6.12 |
| OverJointMA | ShadowHandOver_Safe_joint | 2 | 422 | 40 | ✓ | 6.58 |

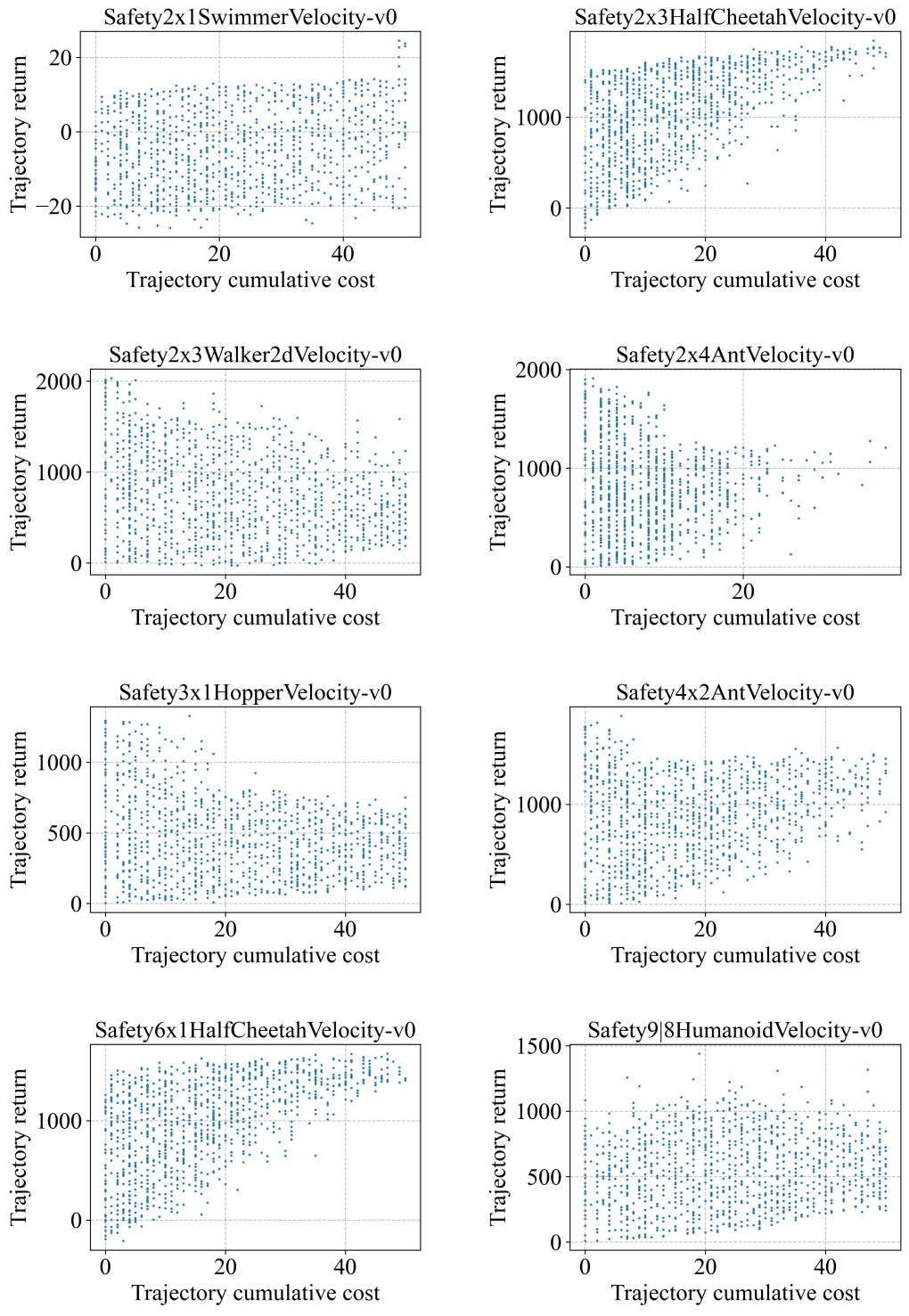

Figure 5: The data distribution over the reward-cost plane of the proposed MOSDB dataset (part 1/3). Each point represents a return-cumulative cost tuple of a trajectory. The cost limit is set to 25.

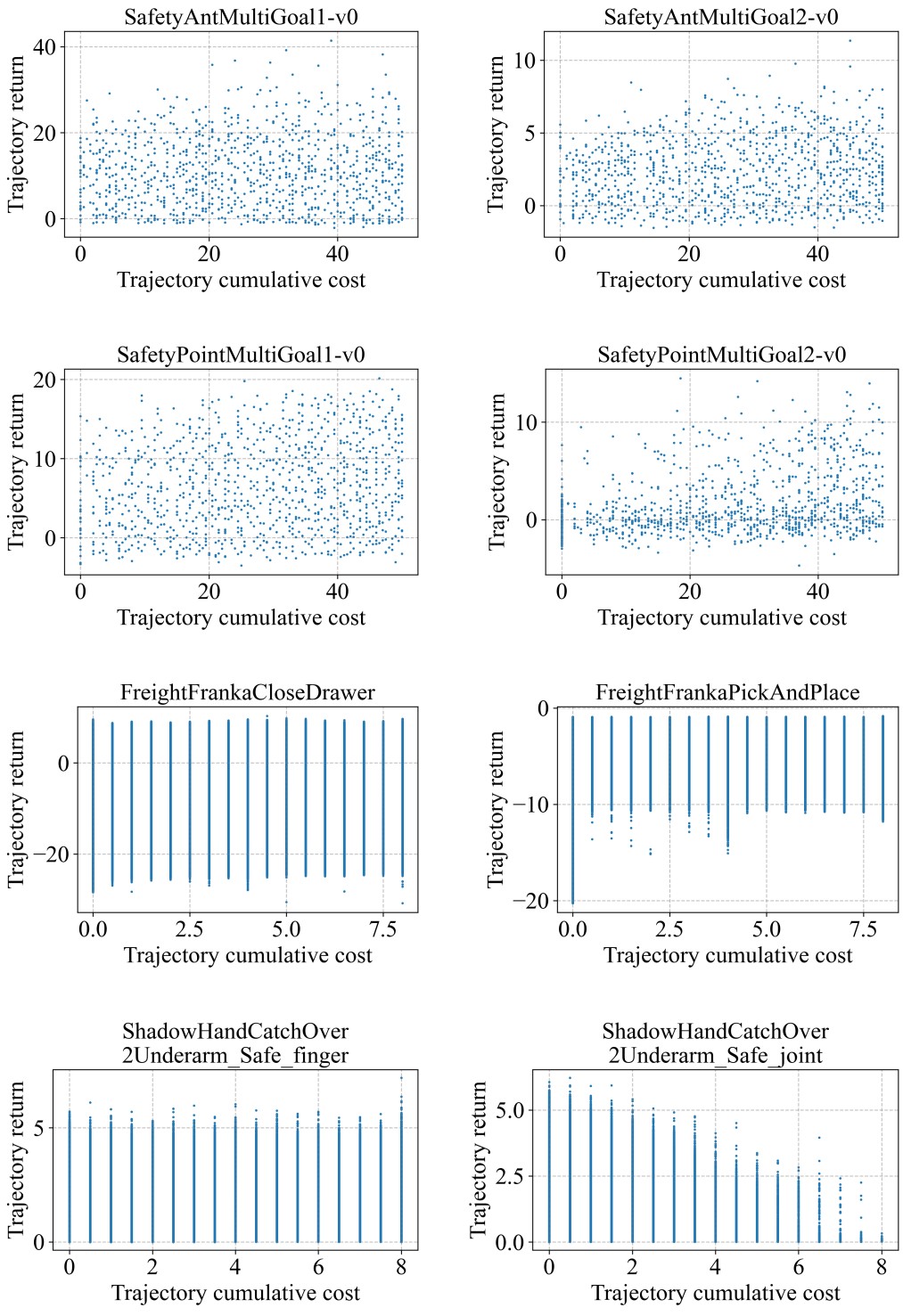

Figure 5: The data distribution over the reward-cost plane of the proposed MOSDB dataset (part 2/3). Each point represents a return-cumulative cost tuple of a trajectory. The cost limit is set to 25.

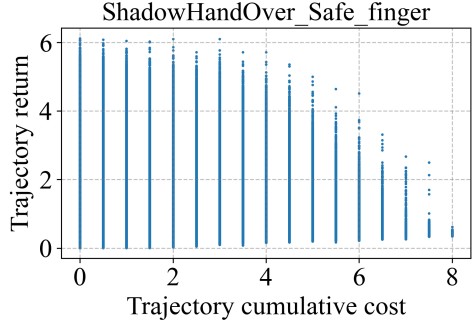
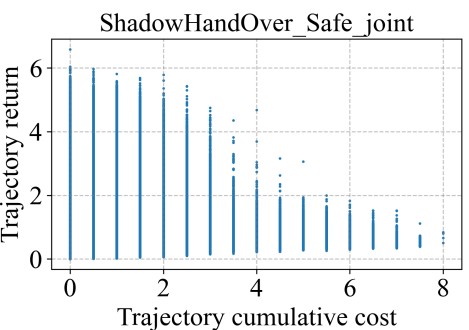

Figure 5: The data distribution over the reward-cost plane of the proposed MOSDB dataset (part 3/3). Each point represents a return-cumulative cost tuple of a trajectory. The cost limit is set to 25.

