# OpenReview forum: "MOSDT: Self-Distillation-Based Decision Transformer for Multi-Agent Offline Safe Reinforcement Learning"
_NeurIPS.cc/2025/Conference — NeurIPS 2025 poster_

### Official Review · Reviewer_eJKo · 2025-06-22

**Clarity:** 2
**Significance:** 2
**Originality:** 2
**Rating:** 4
**Confidence:** 4

**Summary:**

The paper proposes MOSDT, an algorithm for multi-agent offline safe reinforcement learning (MOSRL), and introduces MOSDB, a corresponding benchmark and dataset. The method incorporates policy self-distillation with a global information reconstruction mechanism that fuses observation features across agents. It employs full parameter sharing and introduces a cost binary embedding (CBE) module to encode safety constraints. Empirical results on MOSDB, which includes MuJoCo, Safety Gym, and Isaac Gym environments, show that MOSDT achieves higher returns on 14 out of 18 tasks, while maintaining safety and reducing execution-time parameter count compared to a single-agent baseline.

**Questions:**

Please refer to the weakness.

**Ethical Concerns:**

["NO or VERY MINOR ethics concerns only"]

**Final Justification:**

Given authors' response, I change the scores accordingly.

**Limitations:**

Yes

**Quality:**

2

**Strengths And Weaknesses:**

# Strengths
- The authors extend multi-agent offline reinforcement learning and safe reinforcement learning to the underexplored setting of multi-agent safe offline reinforcement learning.
- They provide a benchmark and dataset tailored to this new problem formulation.
- The proposed approach achieves state-of-the-art performance under safety-constrained evaluation scenarios. A comprehensive ablation study offers insight into the rationale behind key design choices.

# Weaknesses
- There is room for improvement in the paper’s writing. For example, figure captions currently provide limited information and could be more descriptive to aid understanding. In Line 147, the symbol L appears to denote the loss, whereas Equation (1) uses the term Loss. This inconsistency in notation could be clarified.
- While full parameter sharing across agents helps stabilize training, it may limit the applicability of the proposed method in scenarios where agents have heterogeneous action spaces.
- Although the proposed cost binary embedding (CBE) module appears to improve information aggregation and support specific cost targets, its impact on training stability is unclear. It would be helpful if the authors could provide learning curves or additional ablation results to better assess the influence of CBE on training dynamics.

---

> ### Author Rebuttal · Authors · 2025-07-31
>
> We appreciate Reviewer eJKo for recognizing the significance, quality, and completeness of our work. Below, we respond to the weaknesses.
>
> **(Weakness 1) Writing improvements**
>
> We appreciate Reviewer eJKo’s feedback on the writing quality. In the revised version, we will expand the captions to include more information and explanation.
>
> (1) Figure 1: Training MOSDT on the MOSDB dataset. The arrows represent the sequence of operations and data flow among the dataset, the student networks, and the teacher network under the proposed policy self-distillation (PSD) framework.
>
> (2) Figure 2: The network structure of the proposed MOSDT algorithm. The purple areas indicate our proposed innovative modules. The green areas represent data, while the blue areas represent networks. We enlarge one of the DSSL modules and display it in detail in the upper left corner. We only illustrate the situation at time t.
>
> (3) Figure 3: Number of parameters. “Using KD”: Using conventional KD instead of PSD. “No PS”: No parameter sharing. MOSDT only requires 65% of the execution parameter count of CDT. PSD reduces the number of training parameters by 47% and the number of execution parameters by 58%.
>
> (4) Figure 4: Visualization of the training on the “3x1Hopper” task. “No PS”: No parameter sharing. The purple curve in Subgraph (a) is enlarged and displayed in detail in Subgraph (b) for better viewing. Shaded areas represent sample standard deviations across multiple runs.
>
> (5) Figure 5: The data distribution over the reward-cost plane of the proposed MOSDB dataset. Each point represents a return-cumulative cost tuple of a trajectory. The cost limit is set to 25.
>
> Regarding the notation clarity, we confirm that the “L” in line 147 denotes the training sequence length instead of loss. We acknowledge that “L” could indeed be misunderstood as loss, so we will replace it with another notation, such as “M”, in the revised version. We will further check the clarity of notations to eliminate any possible misunderstandings.
>
> **(Weakness 2) Limitations of the full parameter sharing design**
>
> For agents with heterogeneous action spaces, full parameter sharing can effectively stabilize training and improve performance at the scale of the MOSDB dataset (up to 756 dimensions of observation spaces and 52 dimensions of action spaces). Full parameter sharing improves returns on 11 out of 18 tasks compared with no parameter sharing (cf. Table 2) and 12 to 14 out of 18 tasks compared with partial parameter sharing (cf. Table 8 in Appendix B.4).
>
> For all tasks in the MOSDB dataset, agents have heterogeneous action spaces.
>
> (1) MOS Velocity. Agents need to control distinct robot segments to move as quickly as possible while adhering to velocity constraints. The agents are heterogeneous because they play different roles in movements (e.g., the front and rear legs of “HalfCheetah” need to have different behavior patterns to move forward).
>
> (2) MOS Goal. Multiple agents are required to navigate to targets matching their designated colors, while evading collisions and hazardous terrain. The agents are heterogeneous because they have different objectives (navigating to targets in different colors) and do not share rewards and costs (cf. Table 13 in Appendix C), which makes them not interchangeable.
>
> (3) MOS Isaac Gym. It focuses on collaborative tasks between robotic agents, such as dual manipulators performing coordinated object manipulation like ball-handovers. Agents are heterogeneous because they play different roles in collaborations (e.g., dual manipulators are responsible for throwing and catching the ball, respectively).
>
> To verify the effectiveness of full parameter sharing in more challenging scenarios with agents having heterogeneous action spaces, we will introduce a series of higher-dimensional power grid scheduling tasks to future versions of the MOSDB dataset.
>
> **(Weakness 3) Impact of CBE on training dynamics**
>
> We appreciate Reviewer eJKo for the suggestion to analyze the effect of CBE on training dynamics. According to the uploaded learning curves (cf. the link in the abstract), CBE helps improve training stability on almost all tasks by eliminating the noise in cost information. For training dynamics, CBE provides better training initializations and safety controls on most tasks.
>
> The following statements need to be read in conjunction with the uploaded learning curves (cf. the link in our abstract). We observe three main advantages of CBE on training dynamics.
>
> (1) Stabilizing training. On almost all tasks (e.g., “2x3HalfCheetah”, “2x3Walker2d”, and “3x1Hopper”), CBE results in smoother learning curves, compared with MOSDT without CBE. CBE mitigates volatility and reduces sharp declines. The cost binarization in CBE eliminates the noise in the cost information, thereby stabilizing the training.
>
> (2) Better training initialization. In most tasks (e.g., “2x4Ant”, “4x2Ant”, and “Multi-Point1”), CBE yields higher returns with safe cumulative costs at the beginning of training. CBE directly transmits the prior information about the reward-cost correlation to MOSDT by embedding safety signals into returns, enabling agents to learn more efficiently. The cost binarization in CBE eliminates the noise of cost data, improving data efficiency, thereby offering a better training initialization.
>
> (3) Better safety control. In most tasks (e.g., “2x1Swimmer”, “2x4Ant”, and “OverFingerMA”), CBE yields cumulative cost learning curves with faster and more stable convergence. For 11 out of 18 tasks, CBE reduces the cumulative costs (cf. the ablation study results in Table 2). CBE provides better safety control by improving the processing of cost information.
>
> We will add the above analysis and related learning curves to the revised version.

---

> > ### Comment · Reviewer_eJKo · 2025-08-05
> >
> > Thanks the authors for the additional experiments and clarifications on my question. I changed the score accordingly.

---

> > > ### Author Response · Authors · 2025-08-06
> > > **Thank you for your follow-up and for updating the score**
> > >
> > > Dear Reviewer eJKo,
> > >
> > > Thank you for your valuable follow-up and for updating your score. We appreciate your recognition that most of the concerns have been addressed. Your comments has been very helpful in refining our work. Please let us know if you need any further information or if there are additional points you would like to discuss with us. Wishing you all the best in your professional and personal endeavors!
> > >
> > > Best regards,
> > >
> > > Authors of Paper Submission #4775

---

### Official Review · Reviewer_mXFE · 2025-07-01

**Clarity:** 3
**Significance:** 3
**Originality:** 3
**Rating:** 5
**Confidence:** 4

**Summary:**

This paper studies the setting of multi-agent offline safety reinforcement learning. They proposed the first algorithm named MOSDT in this setting and showed its effectiveness through experiments. Besides, they constructed the first dataset and benchmark for this setting. They build their algorithm based on constrained decision transformer (safety offline RL) and centralized training with decentralized execution framework (multi-agent offline RL). Their innovations are from three perspectives. First, they proposed a cost binary embedding module to convert the safety signal to binary signal which solves the issue of unsafe intermediate actions. Second, they introduced self-distillation technique to multi-agent offline RL which simplifies the training pipeline, decreases training parameters and saves the training time. Last, they propose a full-parameter sharing design across students to stabilize the training.

**Questions:**

1. Can the authors explain more on the full parameter sharing design? Based on my understanding, it is to initialize the network of all student agents with the same parameters before the training pipeline starts.

**Ethical Concerns:**

["NO or VERY MINOR ethics concerns only"]

**Final Justification:**

I am recommending an accept since their proposed method is reasonable under the studied setting and the experimental results are satisfactory. But I agree the novelty is limited somehow. Basically, their proposed method is a combination of existing modules (CDT and CTDE) plus some small new tricks (full-parameter sharing and CBE).

**Limitations:**

Yes.

**Paper Formatting Concerns:**

No.

**Quality:**

3

**Strengths And Weaknesses:**

Strengths:
1. They proposed the first algorithm for the setting of multi-agent offline safety reinforcement learning which initiates the study in this setting.
2. The proposed techniques including cost binary embedding and self-distillation look reasonable to me for this problem.
3. They did comprehensive ablation studies to show the effect of different components in their algorithm network.

Weaknesses:
Not much. The self-distillation technique is not tailored to safety RL which means it can be applied to multi-agent offline RL without safety constraints. Would be interesting to see how it works outside the safety setting compared to works such as MADTKD. The experiments are all based in the safety setting.

---

> ### Author Rebuttal · Authors · 2025-07-31
>
> We appreciate Reviewer mXFE for recognizing the significance, reasonability, and completeness of our work. Below, we respond to the weakness and the question.
>
> **(Weakness 1) Effectiveness of policy self-distillation (PSD) in offline MARL**
>
> We appreciate the Reviewer mXFE for the suggestion to analyze PSD without safety settings. To verify the effectiveness of PSD in offline MARL without safety constraints, we removed the safety constraints in the environments of the MOSDB benchmark and let agents run 1,000 steps during evaluation. In this setting, we train MOSDT with the proposed PSD and with the conventional knowledge distillation (used in MADTKD [1]), respectively. The results are shown in the tables below (with the same setting as Table 2).
>
> Table A: Results of training MOSDT with PSD on the MOSDB benchmark (without safety constraints).
>
> |  |  |  |  |  |  |  |  |  |  |
> |---|---|---|---|---|---|---|---|---|---|
> | Task No. | 1 | 2 | 3 | 4 | 5 | 6 | 7 | 8 | 9  |
> | Return | 13.70 | 2050.09 | 1783.18  | 2171.59  | 1184.62  | 2053.42  | 1885.41 | 1126.75  | 37.90  |
> | Task No. | 10 | 11 | 12 | 13 | 14 | 15 | 16 | 17 | 18  |
> | Return  | 5.01 | 11.38 | 5.06 | 1.14 | -2.87 | 2.59 | 2.24 | 3.11  | 2.92  |
>
> Table B: Results of training MOSDT with the conventional knowledge distillation on the MOSDB benchmark (without safety constraints).
>
> |  |  |  |  |  |  |  |  |  |  |
> |---|---|---|---|---|---|---|---|---|---|
> | Task No. | 1 | 2 | 3 | 4 | 5 | 6 | 7 | 8 | 9  |
> | Return  | ↓12.53 | ↓2027.44 | 1802.80 | ↓2150.33 | ↓1169.34 | 2118.23 | ↓1850.15 | 1208.25 | ↓37.07  |
> | Task No. | 10 | 11 | 12 | 13 | 14 | 15 | 16 | 17 | 18  |
> | Return  | ↓4.45 | 11.51 | 9.86 | ↓-0.02 | -1.97 | ↓2.00 | ↓1.93 | ↓1.28 | ↓2.81  |
>
> According to these two tables, PSD improves returns on 12 out of 18 tasks (without safety constraints) compared to the conventional knowledge distillation. PSD improves most offline MARL tasks (with or without safety settings).
>
> The primary focus of our paper is to tackle the novel and underexplored problem of multi-agent offline safe RL (MOSRL). Offline MARL methods, such as MADTKD [1], cannot address safety constraints, making them unsuitable for direct comparison within our safety-focused experimental framework. Our experiments thus prioritize the safety setting to demonstrate the risk control capabilities of MOSDT. We will conduct more in-depth research into the effectiveness of PSD in offline MARL without safety settings in subsequent work.
>
> **(Question 1) Clarification on the full parameter sharing design**
>
> We thank Reviewer mXFE’s interest in our full parameter sharing design. We note that full parameter sharing is not merely initializing all student networks with the same parameters before training. We share all parameters across all student networks throughout the training process, as described in Eq. (6). This design keeps the training stable during the entire process and significantly reduces model parameters. In practice, we only create one instance of the Student class for all agents. In the entire MOSDT network, only the teacher action heads and the teacher feature projectors are distinct for different agents. According to the ablation study, compared with no/partial parameter sharing, full parameter sharing yields the best performance (cf. Table 2 in Section 4.2 and Table 8 in Appendix B.4) and slashes parameter count to the greatest extent.
>
> In the revised version, we will enhance the description of full parameter sharing. We appreciate Reviewer mXFE for highlighting this point, which allows us to improve the clarity of the paper.
>
> **References**
>
> [1] Tseng W C, Wang T H J, Lin Y C, et al. Offline multi-agent reinforcement learning with knowledge distillation[J]. Advances in Neural Information Processing Systems, 2022, 35: 226-237.

---

> > ### Comment · Reviewer_mXFE · 2025-08-05
> >
> > Thanks the authors for the additional experiments and clarifications on my question. I'll keep my score.

---

> > > ### Author Response · Authors · 2025-08-06
> > > **Thank you for your feedback and support**
> > >
> > > Dear Reviewer mXFE,
> > >
> > > Thank you for your valuable feedback and support. We appreciate your recognition that most of the concerns have been addressed. Your review has been very helpful in refining our work. Please let us know if you need any further information or if there are additional points you would like to discuss with us. Wishing you all the best in your professional and personal endeavors!
> > >
> > > Best regards,
> > >
> > > Authors of Paper Submission #4775

---

### Official Review · Reviewer_vSoS · 2025-07-03

**Clarity:** 3
**Significance:** 3
**Originality:** 3
**Rating:** 4
**Confidence:** 3

**Summary:**

This paper introduces MOSDT, a decision transformer architecture for multi-agent offline safe reinforcement learning. The method consists of several points of this approach, including (1) policy self-distillation integrated with global information reconstruction, (2) full parameter sharing across agents to stabilize training and reduce model complexity, and (3) cost binary embedding that replaces specific cost targets with binary safety signals for more flexible safety handling. The authors also release MOSDB, the first benchmark dataset for MOSRL, and show that MOSDT achieves SOTA performance on 14 out of 18 tasks, ensuring complete safety while requiring only 65% of the execution parameter count compared to baselines.

**Questions:**

Can the authors provide a deeper analysis of the failure cases observed in the ablation study (e.g., the four tasks where safety constraints were violated)? What factors contributed to these violations?

**Ethical Concerns:**

["NO or VERY MINOR ethics concerns only"]

**Final Justification:**

Thank the authors for detailed replies. It addresses my concerns. After reviewing the replies of the authors to me and other reviewers, I would like to keep my score.

**Limitations:**

yes

**Quality:**

3

**Strengths And Weaknesses:**

Strengths:
- This paper addresses the understudied and challenging domain of multi-agent offline safe reinforcement learning, where issues of scalability and safety remain largely unresolved.
- On the new MOSDB benchmark, MOSDT achieves state-of-the-art (SOTA) performance on 14 out of 18 tasks, spanning diverse environments such as MuJoCo, Safety Gym, and Isaac Gym, while strictly adhering to safety constraints.
- Full parameter sharing reduces the execution parameter count by 58% and improves inference time compared to CDT.

Weaknesses:
- Limited real-world baseline comparison: The evaluation primarily adapts single-agent baselines to the multi-agent setting. Strong multi-agent baselines—such as recent diffusion-based methods (e.g., MADiff) or model-based safe MARL algorithms (e.g., constraint-aware MBRL)—are not included. These could offer valuable comparisons, especially under tight safety budgets or dynamic constraints.
- Cost binarization limitation: The proposed CBE encodes safety using hard binary thresholds, which may limit adaptability in settings with soft, dynamic, or probabilistic constraints.
- Limited safety analysis: Although all reported tasks satisfy safety thresholds, the failure modes (e.g., the four unsafe cases in the ablation study) are not thoroughly analyzed. Deeper investigation into these cases could inform future improvements in robustness.

---

> ### Author Rebuttal · Authors · 2025-07-31
>
> We appreciate Reviewer vSoS for recognizing the significance and quality of our work. Below, we respond to the weakness and the question.
>
> **(Weakness 1) Limited baseline comparison**
>
> We appreciate the Reviewer vSoS’s suggestion to broaden our evaluation by incorporating advanced offline MARL methods. We train MADiff [1] (a diffusion-based offline MARL method) and MOMA-PPO [2] (a model-based offline MARL method) on the proposed MOSDB dataset with their default settings for the tasks with the highest state dimensions. To make these two methods applicable to safety constraints, we input cumulative costs into them along with returns. The results are shown in the tables below (with the same setting as Table 1).
>
> Table A: Results of training MADiff [1] on the MOSDB dataset.
>
> |  |  |  |  |  |  |  |  |  |  |
> |---|---|---|---|---|---|---|---|---|---|
> | Task No. | 1 | 2 | 3 | 4 | 5 | 6 | 7 | 8 | 9  |
> | Return (cost)  | ↓8.29 (18.40) | 2131.33 (55.23, unsafe) | ↓1560.67 (0.63) | 2197.03 (2.47) | ↓81.67 (8.57) | ↓880.59 (0.00) | ↓357.98 (0.00) | 613.12 (19.93) | ↓20.44 (18.17)  |
> | Task No. | 10 | 11 | 12 | 13 | 14 | 15 | 16 | 17 | 18  |
> | Return (cost) | ↓2.20 (19.28) | ↓3.50 (7.00) | ↓-3.35 (21.47) | ↓-5.19 (1.00) | ↓-3.95 (0.00) | ↓0.24 (0.00) | ↓0.20 (0.00 | ↓0.44 (0.00) | ↓0.45 (0.07)  |
>
> Table B: Results of training MOMA-PPO [2] on the MOSDB dataset.
>
> |  |  |  |  |  |  |  |  |  |  |
> |---|---|---|---|---|---|---|---|---|---|
> | Task No. | 1 | 2 | 3 | 4 | 5 | 6 | 7 | 8 | 9  |
> | Return (cost) | ↓8.84 (19.23) | ↓665.63 (10.67) | ↓1557.61 (1.63) | 2296.26 (5.17) | ↓31.33 (0.00) | ↓734.56 (0.00) | ↓335.55 (0.00) | 552.37 (19.50) | ↓17.01 (19.97)  |
> | Task No. | 10 | 11 | 12 | 13 | 14 | 15 | 16 | 17 | 18  |
> | Return (cost) | ↓2.22 (16.50) | ↓0.32 (24.78) | 0.59 (15.17) | ↓-5.33 (1.00) | ↓-4.26 (0.00) | ↓0.24 (1.33) | ↓0.21 (0.00) | ↓0.45 (0.03)  | 0.47 (0.47)  |
>
> Compared with our proposed MOSDT, MADiff [1] yields lower returns on 15 out of 18 tasks and leads to unsafe policy on 1 task (Task No. 2). MOMA-PPO [2] yields lower returns on 14 out of 18 tasks compared with MOSDT, while maintaining safety. We will supplement these new results in the revised version and upload their training curves.
>
> Although advanced offline MARL methods like MADiff [1] and MOMA-PPO [2] perform well in standard offline MARL settings, they lack safety-oriented designs, limiting their applicability to safety-critical MOSRL tasks.
>
> **(Weakness 2) Limitations of CBE**
>
> We acknowledge that the cost binarization in CBE may limit its adaptability in settings with soft, dynamic, or probabilistic constraints, as stated in Appendix D (lines 520-522). However, in most cases, the cost limits are fixed values, such as the temperature limits of cables in power systems [3] and the distance limit between vehicles and human in autonomous driving [4]. Many widely used safe RL benchmarks, such as Gymnasium [5] and DSRL [6], also adopt fixed values as their cost limits. In most cases where the cost limits are fixed, CBE offers a better alternative to specific cumulative costs, given its powerful effect (improving returns on 14 out of 18 tasks in Table 2), plug-and-play feature, and intuitive design.
>
> In the future work, we will exploration several directions to improve our approach to accommodate soft, dynamic, or probabilistic constraints, enhancing its adaptability to more complex and realistic settings.
>
> (1) Soft safety constraints: Replacing hard binary signals with a continuous cost representation. This would allow the model to reason about the severity of safety violations rather than treating them as all-or-nothing events, improving performance in environments with graded safety settings.
>
> (2) Dynamic safety constraints: Introducing mechanisms to adjust thresholds based on contextual factors or trajectory history. This could enhance flexibility in settings where safety requirements evolve.
>
> (3) Probabilistic safety constraints: Integrating uncertainty-aware models to handle probabilistic safety constraints, which could improve performance in applications where safety requirements are not deterministic.
>
> **(Weakness 3 & Question 1) Analysis of failure cases**
>
> We appreciate Reviewer vSoS’s suggestion on analyzing the failure cases without safety constraints violated. There are three major causes for the violation of safety constraints.
>
> (1) High risk in the early stage of training. As shown in the uploaded training curves (cf. the link in the abstract), most failure cases show excessive cumulative costs at the initial stages of training. For example, on the “2x3HalfCheetah” task (the cost limit is 25), when 10% of the training (10k steps) is completed, the complete MOSDT only yields a cumulative cost of 13.1, while the MOSDT without PSD yields a cumulative cost of 18.8, and CDT [7] yields a cumulative cost of 62, resulting the violation of safety constraints of the latter two methods (cf. Tables 1 and 2).
>
> (2) Instability of the cumulative costs in training. In most failure cases, the training curves of cumulative costs show severe fluctuations (e.g., the training curves of BCQ-Lag [8] on the “Multi-Point2” task). This instability indicates that the model often enters unsafe states and cannot maintain safety.
>
> (3) Lack of communication among agents. Maintaining global safety requires cooperation among agents. Lack of communication among agents could cause them to violate safety constraints. For example, in the "2x3HalfCheetah" task, rollout visualizations of the MOSDT without PSD (i.e., lacking centralized training) revealed that the agent's hind leg persistently accelerated, while its front leg remained stationary to decelerate. This uncoordinated action led to over-speeding and eventual task failure (cf. Table 2).
>
> We will supplement the above analysis in the revised version.
>
> **References**
>
> [1] Zhu Z, Liu M, Mao L, et al. Madiff: Offline multi-agent learning with diffusion models[J]. Advances in Neural Information Processing Systems, 2024, 37: 4177-4206.
>
> [2] Barde P, Foerster J, Nowrouzezahrai D, et al. A model-based solution to the offline multi-agent reinforcement learning coordination problem[J]. arXiv preprint arXiv:2305.17198, 2023.
>
> [3] Chen X, Qu G, Tang Y, et al. Reinforcement learning for selective key applications in power systems: Recent advances and future challenges[J]. IEEE Transactions on Smart Grid, 2022, 13(4): 2935-2958.
>
> [4] Chen Y, Ji C, Cai Y, et al. Deep reinforcement learning in autonomous car path planning and control: A survey[J]. arXiv preprint arXiv:2404.00340, 2024.
>
> [5] Ji J, Zhang B, Zhou J, et al. Safety gymnasium: A unified safe reinforcement learning benchmark[J]. Advances in Neural Information Processing Systems, 2023, 36: 18964-18993.
>
> [6] Liu Z, Guo Z, Lin H, et al. Datasets and benchmarks for offline safe reinforcement learning[J]. arXiv preprint arXiv:2306.09303, 2023.
>
> [7] Liu Z, Guo Z, Yao Y, et al. Constrained decision transformer for offline safe reinforcement learning[C]//International conference on machine learning. PMLR, 2023: 21611-21630.
>
> [8] Xu H, Zhan X, Zhu X. Constraints penalized q-learning for safe offline reinforcement learning[C]//Proceedings of the AAAI Conference on Artificial Intelligence. 2022, 36(8): 8753-8760.

---

### Official Review · Reviewer_Twb8 · 2025-07-05

**Clarity:** 3
**Significance:** 1
**Originality:** 2
**Rating:** 2
**Confidence:** 4

**Summary:**

This paper investigates multi-agent offline safe reinforcement learning. Built on top of CDT and MADTKD, this paper proposes MOSDT, which performs (1) decentralized student supervised learning; (2) centralized teacher supervised learning; (3) policy self-distillation. Also, this paper utilizes several implementation tricks: (1) full parameter sharing and (2)
Cost binary embedding. Extensive experiments are conducted on a self-collected dataset MOSDB to demonstrate the effectiveness of the proposed algorithm.

**Questions:**

See weakness

**Ethical Concerns:**

["NO or VERY MINOR ethics concerns only"]

**Quality:**

2

**Strengths And Weaknesses:**

### Strengths
1. Comprehensive ablation studies are conducted to verify each component of the proposed algorithm.
2. The techniques are simple and easy to follow.

### Weaknesses

1. The setting seems too incremental for considering offline, multi-agent, safety RL. Please motivate this setting with several important application scenarios. It would be better to include a benchmark that tackles one more practical scenario to make this setting well-justified.

2. The proposed algorithm is also incremental over two existing algorithms, one from offline safety RL algorithm, CDT, and one from offline multi-agent RL algorithm, MADTKD. There is not any clear motivation in the methodology to justify why we should design MOSRL algorithms as in MOSDT.

3. The quality of the built benchmark MOSDB seems low. As shown in Table 1 and Table 11, MOS Goal and MOS Isaac Gym cannot provide informative comparison between MOSRL algorithms, as the performance gap can be filled even with several random seeds. Same phenomenon can be overserved in Table 2 and Table 12. This makes the reported performance based on SOTA on how many settings meaningless. Also, it is weird to report mean values and standard deviations in two separate tables. Please consider merge them in one table.

4. The motivation of the paper is clear. It states that multi-agent offline safe Reinforcement Learning is lacking in the existing literature, but it does not specify why this particular technique is necessary. What specific problem is this paper to address? Why are existing solutions inadequate for this purpose?

5. What is the exact novelty compared to reference [5]?

6.  MOSDB in Section 3.4: What is the nature of the tasks within the datasets? How are the trajectories collected (what is the meaning of the trajectories in the dataset, anyway)? The authors are expected to elaborate further on this newly created dataset, including the motivation, tasks, evaluation methods, baselines, etc.

---

> ### Author Rebuttal · Authors · 2025-07-27
>
> We appreciate Reviewer Twb8 for recognizing the completeness and usability of our work. Below, we respond to the weaknesses.
>
> **(W1) Motivation for multi-agent offline safe RL (MOSRL)**
>
> We further emphasize that MOSRL is not merely an incremental extension but a critical advancement addressing challenges in many important application scenarios.
>
> (1) Autonomous vehicle coordination [1]. Driving policies for autonomous vehicles should be safe and cooperative to avoid any collisions with each other. Online exploration is impractical due to safety risk.
>
> (2) Power grid scheduling [2]. A power grid involves multiple agents (e.g., generators). Safety constraints, such as preventing overload, are crucial. Online trial-and-error is infeasible due to potential catastrophic consequences.
>
> (3) Robot collaboration [3]. In industrial settings such as manufacturing and logistics, multiple robots need to coordinate under strict safety constraints (e.g., collision avoidance during collaborative machining and transportation). Online learning may cause unacceptable asset losses.
>
> We mentioned scenarios (1) and (2) in Section 1 (lines 29-30). We will emphasize more on the motivation of MOSRL in the revised version. The complex requirements in these scenarios reveal the limitations of existing RL methods. The developed MOSRL framework, algorithm, and benchmarks have the potential to meet the requirement of learning safe, cooperative policies from offline data.
>
> Indeed, the MOSDB dataset has incorporated 18 practical tasks, including bionic robot racing, robot navigation and obstacle avoidance, and throwing/catching objects by bionic robotic hands. These tasks have significant implications in various real-world engineering domains, including autonomous vehicles and industrial automation.
>
> We recognize the value in expanding the scope for greater practical significance. As stated in Appendix D (lines 522-523), future efforts will focus on expanding the MOSDB dataset, by introducing a series of power system operation tasks with real-world data.
>
> **(W2) Design motivation of MOSDT**
>
> Instead of a simple combination of CDT and MADTKD, MOSDT is a new algorithm tailored for the MOSRL problem, with several key innovations that significantly enhance its performance in multi-agent environments with safety constraints.
>
> We propose two innovative designs to address key challenges in multi-agent environments, including policy self-distillation (PSD) and full parameter sharing. Inspired by the spirit of MADTKD that introduces policy distillation to offline MARL, we propose PSD to extend CDT to multi-agent settings. Different from MADTKD, PSD embeds student networks into the teacher and performs distillation synchronously with supervised learning, reducing training parameter count by 35% and training time by 24%. As shown in the following table (with the same setting as Table 2), compared with the PSD in our proposed MOSDT, policy distillation (in MADTKD) causes return losses on 11 out of 18 tasks and leads to unsafe policy on 1 task (Task No. 2).
>
> Table A
>
> |  |  |  |  |  |  |  |  |  |  |
> |---|---|---|---|---|---|---|---|---|---|
> | Task No. | 1 | 2 | 3 | 4 | 5 | 6 | 7 | 8 | 9  |
> | Return (cost) | ↓10.56 (13.90) | 2096.33 (39.07, unsafe) | ↓1546.32 (3.30) | ↓2035.09 (4.43) | 1256.79 (10.23) | 2197.64 (5.47) | ↓1808.23 (24.63) | 499.15 (20.47) | ↓31.60 (14.83)  |
> | Task No. | 10 | 11 | 12 | 13 | 14 | 15 | 16 | 17 | 18  |
> | Return (cost) | 3.35 (14.00) | 11.33 (8.33) | 3.32 (16.83) | ↓-3.74 (0.00) | ↓-2.69 (0.00) | ↓0.20 (2.97) | ↓0.24 (0.00) | ↓0.43 (0.80) | ↓0.43 (1.10)  |
>
> The full parameter sharing among agents reduces training parameter count by 47% and execution parameter count by 58% (cf. Fig. 3).
>
> We propose a plug-and-play module, cost binary embedding (CBE), to address key challenges in safety-critical settings. For MOSDT, CBE enhances the cost information processing capabilities of the CDTs by binarizing cumulative costs as safety signals and embedding the signals into returns. CBE improves returns on 14 out of 18 tasks (cf. Table 2).
>
> **(W3) Quality of the MOSDB benchmark**
>
> Our benchmark MOSDB can provide an informative comparison on most of the tasks in a widely adopted environment, *Safety Gymnasium* [4]. Indeed, MOSDT achieves SOTA returns on 14 out of 18 tasks (cf. Table 1). On 9 out of these 14 tasks, MOSDT achieves a lead of more than 5% compared to the suboptimal method. On some tasks, MOSDT achieves a return several times higher than the suboptimal method (9.2 times on “3x1Hopper”, 4.2 times on “6x1HalfCheetah”, and 2.3 times on “4x2Ant”). On 13 out of these 14 tasks, MOSDT achieves a lead of more than 10% compared to the average performance of the baseline models.
>
> A few tasks in the MOSDB benchmark demonstrate small performance gaps among methods—we note that this is not unusual in other well-known benchmarks. For instance, in Safety Gymnasium [4], the best method achieves a lead of more than 5% compared to the suboptimal method on only 2 out of 6 Safety Velocity tasks. We agree with Reviewer Twb8 that for MOSDB, the statistical significance of performance comparison requires more attention.
>
> The offline data in the MOSDB dataset is collected strictly following the procedure of a widely used single-agent offline safe dataset, DSRL [5].  Following existing well-known RL benchmarks, such as Safety Gymnasium [4], DSRL [5], and D4RL [6], we evaluate the quality of algorithms based on achieving SOTA average returns on how many tasks.
>
> In the original submission, we separated mean values and standard deviations due to space constraints. We will merge them into one table in the revised version.
>
> **(W4) Necessity of MOSRL**
>
> As mentioned in Section 1 (lines 29-30), MOSRL is promising in various application scenarios, such as autonomous vehicle coordination and robot collaboration (cf. our reply to W1 for more details), where agents need to learn safe, cooperative policies using offline data.
>
> As stated in Section 1 (lines 21-26), existing offline RL methods focus solely on either multi-agent or safety settings, leaving a gap in algorithms specifically designed for MOSRL. Existing offline MARL methods (e.g., MADT, MADiff, and PTDE) lack built-in safety mechanisms, rendering them ineffective for safety-critical applications. Current offline safe RL methods (e.g., CDT, SaFomer, and SDT) are tailored for single-agent settings, lacking the necessary coordination mechanisms for multi-agent systems.
>
> To lay the foundation for MOSRL, we provide its first algorithm (MOSDT) and first dataset (MOSDB). We train MOSDT and other baseline methods to set up the MOSDB benchmark and verify the superiority of MOSDT in addressing MOSRL challenges.
>
> **(W5) Novelty Compared to MADTKD**
>
> Compared to MADTKD, the proposed MOSDT incorporates three key innovations.
>
> (1) Policy self-distillation (PSD). As a two-stage method, it first trains an independent teacher network and then distills the policy to students. We introduce policy self-distillation (PSD) to MARL. PSD synchronously performs policy distillation alongside supervised policy learning, integrating student networks into the teacher network to streamline training and enhance parameter efficiency.
>
> (2) Full parameter sharing. Existing offline MARL methods (including MADTKD) assign distinct network parameters to different agents. We propose full parameter sharing across all agents to ensure training stability and make MOSDT more lightweight.
>
> (3) Safety property with CBE. As an offline MARL algorithm, MADTKD was not designed to address safety constraints. To enhance the safety performance of MOSDT, we propose a new plug-and-play module, CBE, which i) binarizes cumulative costs as safety signals, and ii) embeds the signals into returns.
>
> **(W6) Details of the MOSDB Dataset**
>
> The offline data in the MOSDB dataset is collected from Safety Gymnasium [4]. Please refer to [4] for detailed introduction on the nature of the tasks. We briefly describe these tasks below and will supplement this information in the revised version.
>
> (1) MOS Velocity. Robots are required to move as quickly as possible while adhering to velocity constraints. Multiple agents need to control distinct body segments cooperatively.
>
> (2) MOS Goal. Each agent is required to reach its color-designated target while avoiding collisions and hazardous terrain.
>
> (3) MOS Isaac Gym. It focuses on collaborative robotic tasks, such as coordinated ball-handovers between dual manipulators, with enforced safety constraints on joint movements.
>
> Regarding trajectory collection, as stated in Section 1 (line 77), we train two multi-agent safe RL algorithms on Safety Gymnasium [4], logging agent-environment interactions as offline data. As a commonly used concept in offline RL, “trajectory” is a fixed sequence of historical state-action-reward (cost) transitions.
>
> As mentioned in Section 2 (line 114), the lack of MOSRL datasets is our key motivation for building the MOSDB dataset. The evaluation methods are described in Section 4.1 (lines 241-246). The baseline methods trained on the MOSDB dataset are introduced in Section 4.1 (lines 232-239).
>
> We will integrate some of the above details of MOSDB into Section 3.4 in the revised version.
>
> **References**
>
> [1] Chen Y, et al. Deep reinforcement learning in autonomous car path planning and control: A survey. arXiv (2024).
>
> [2] Chen X, et al. Reinforcement learning for selective key applications in power systems: Recent advances and future challenges. IEEE Transactions on Smart Grid (2022).
>
> [3] Singh B, et al. Reinforcement learning in robotic applications: a comprehensive survey. Artificial Intelligence Review (2022).
>
> [4] Ji J, et al. Safety gymnasium: A unified safe reinforcement learning benchmark. NeurIPS (2023).
>
> [5] Liu Z, et al. Datasets and benchmarks for offline safe reinforcement learning. arXiv (2023).
>
> [6] Fu J, et al. D4rl: Datasets for deep data-driven reinforcement learning. arXiv (2020).

---

> ### Author Response · Authors · 2025-08-06
> **Looking forward to your feedback**
>
> Dear Reviewer Twb8,
>
> We hope that our rebuttal has effectively addressed all your concerns. As the deadline for the discussion period draws near, we would greatly appreciate it if you could re-evaluate our manuscript and provide any feedback. Thank you for your time and consideration.
>
> Best regards,
>
> Authors of Paper Submission #4775

---

> ### Author Response · Authors · 2025-08-08
> **Looking forward to your feedback**
>
> Dear Reviewer Twb8,
>
> We hope that our rebuttal has effectively addressed all your concerns. As the deadline for the discussion period draws near (**less than 24 hours**), we would greatly appreciate it if you could re-evaluate our manuscript and provide any feedback. Thank you for your time and consideration.
>
> Best regards,
>
> Authors of Paper Submission #4775

---

> ### Author Response · Authors · 2025-08-09
> **Looking forward to your feedback (urgent)**
>
> Dear Reviewer Twb8,
>
> We hope that our rebuttal has effectively addressed all your concerns. As the deadline for the discussion period draws very near (**less than only 1 hour**), we would greatly appreciate it if you could re-evaluate our manuscript and provide any feedback. Thank you for your time and consideration.
>
> Best regards,
>
> Authors of Paper Submission #4775

---

### Author Response · Authors · 2025-08-09
**Rebuttal Summary**

We are grateful to all the reviewers for their time and insightful feedback. **We appreciate the reviewers‘ recognition of the significance of our contribution to a new RL subfield (Reviewers eJKo, vSoS, and mXFE), the completeness of our experiments (Reviewers Twb8, eJKo, and mXFE), and the quality of our proposed algorithm and benchmark (Reviewers eJKo and vSoS).**

We also sincerely appreciate the concerns some reviewers have raised regarding the key contributions and the uniqueness of our manuscript. We appreciate the opportunity to clarify these points. **We propose the first algorithm (MOSDT) and the first benchmark (MOSDB) for multi-agent offline safe RL (MOSRL)**, a new RL subfield with significant potential for distributed safety-critical applications like vehicle coordination and grid scheduling. **The proposed MOSDT incorporates three key innovations**: i) PSD, streamlining the training process and the network structure, marking the first demonstration that self-distillation is effective in MARL, ii) full parameter sharing, ensuring training stability and making MOSDT more lightweight, and iii) CBE, a new plug-and-play module that enhances cost information processing. **On the MOSDB benchmark, MOSDT achieves SOTA performance on 14 out of 18 tasks with only 65% of the execution parameter count of our base model.**

**We have completed all further experiments and analyses requested by reviewers’ comments**, including weaknesses 1 and 3 raised by Reviewer vSoS, weakness 1 raised by Reviewer mXFE, and weakness 3 raised by Reviewer eJKo. **All supplementary results further support the conclusions in our paper. We have addressed all of the reviewers' concerns especially those regarding our algorithm and benchmark**, including the quality of the proposed MOSDB benchmark (weakness 3 raised by Reviewer Twb8), the limitations of the CBE module (weakness 2 raised by Reviewer vSoS), and the limitations of the full parameter sharing design (weakness 2 raised by Reviewer eJKo). We will include these results in the revised version, making our paper more complete.

Once again, we would like to express our most sincere gratitude to all Reviewers, Program Chairs, Senior Area Chairs, and Area Chairs for the valuable evaluations and assistance regarding our submission. We sincerely hope our paper can receive objective and proper evaluations. Wishing you all the best!

---

### Note · Authors · 2025-08-14

**Rebuttal summary**

We are grateful to all the reviewers for their time and insightful feedback. **We appreciate the reviewers' recognition of the significance of our contribution to a new RL subfield (Reviewers eJKo, vSoS, and mXFE), the completeness of our experiments (Reviewers Twb8, eJKo, and mXFE), and the quality of our proposed algorithm and benchmark (Reviewers eJKo and vSoS).**

We also sincerely appreciate the concerns some reviewers have raised regarding the key contributions and the uniqueness of our manuscript. We appreciate the opportunity to clarify these points. **We propose the first algorithm (MOSDT) and the first benchmark (MOSDB) for multi-agent offline safe RL (MOSRL)**, a new RL subfield with significant potential for distributed safety-critical applications like vehicle coordination and grid scheduling. **The proposed MOSDT incorporates three key innovations**: i) PSD, streamlining the training process and the network structure, marking the first demonstration that self-distillation is effective in MARL, ii) full parameter sharing, ensuring training stability and making MOSDT more lightweight, and iii) CBE, a new plug-and-play module that enhances cost information processing. **On the MOSDB benchmark, MOSDT achieves SOTA performance on 14 out of 18 tasks with only 65% of the execution parameter count of our base model.**

**We have completed all further experiments and analyses requested by reviewers' comments**, including weaknesses 1 and 3 raised by Reviewer vSoS, weakness 1 raised by Reviewer mXFE, and weakness 3 raised by Reviewer eJKo. **All supplementary results further support the conclusions in our paper. We have addressed all of the reviewers' concerns especially those regarding our algorithm and benchmark**, including the quality of the proposed MOSDB benchmark (weakness 3 raised by Reviewer Twb8), the limitations of the CBE module (weakness 2 raised by Reviewer vSoS), and the limitations of the full parameter sharing design (weakness 2 raised by Reviewer eJKo). **We will include these new results in the revised version**, making our paper more complete.

Once again, we would like to express our most sincere gratitude to all Reviewers, Program Chairs, Senior Area Chairs, and Area Chairs for the valuable evaluations and assistance regarding our submission. We sincerely hope our paper can receive objective and proper evaluations. Wishing you all the best!

---

### Decision · Program_Chairs · 2025-09-17

**Decision:**

Accept (poster)

**Comment:**

This paper addresses a novel problem formulation that combines aspects of MARL and OSRL in a non-trivial way. While the solution draws on ideas that are already present in the community, the way they are brought together is new, and the approach demonstrates clear empirical benefits. The formulation itself is interesting and contributes to expanding the scope of problems considered in this area.

The reviewers raised several concerns, most notably regarding the degree of novelty given that the method integrates existing modules (e.g., CDT, CTDE) along with smaller modifications (e.g., parameter sharing, CBE). Some reviewers questioned whether this rises to the level of a NeurIPS contribution. There were also a number of questions around the problem setting and empirical results.

The authors provided thoughtful and detailed responses to these points, which help clarify the contribution. In particular, the rebuttal makes a stronger case for the non-triviality of the problem formulation and the benefits of the proposed design choices.

Overall, this is a promising paper that presents an interesting formulation, demonstrates empirical gains, and helps expand an active research area. With revisions to better highlight the problem’s novelty and the rationale behind the design choices, the paper will be stronger.